# 3,6'-dithiopomalidomide reduces neural loss, inflammation, behavioral deficits in brain injury and microglial activation

**Chih-Tung Lin[1†], Daniela Lecca[2†], Ling-Yu Yang[1†], Weiming Luo[2†], Michael T Scerba[2], David Tweedie[2], Pen-Sen Huang[1], Yoo-Jin Jung[2], Dong Seok Kim[2,3,4], Chih-Hao Yang[5], Barry J Hoffer[6], Jia-Yi Wang[1,7,8]\*, Nigel H Greig[2]\***

[1]Graduate Institute of Medical Sciences, Taipei Medical University, Taipei, Taiwan; [2]Drug Design & Development Section, Translational Gerontology Branch, Intramural Research Program National Institute on Aging, NIH, Baltimore, United States; [3]AevisBio Inc, Gaithersburg, United States; [4]AevisBio Inc, Daejeon, Republic of Korea; [5]Department of Pharmacology, School of Medicine, College of Medicine, Taipei Medical University, Taipei, Taiwan; [6]Department of Neurological Surgery, Case Western Reserve University, Cleveland, United States; [7]Department of Neurosurgery, Taipei Medical University Hospital, Taipei Medical University, Taipei, Taiwan; [8]Neuroscience Research Center, Taipei Medical University, Taipei, Taiwan

**\*For correspondence:**
jywang2010@tmu.edu.tw (J-YW);
Greign@grc.nia.nih.gov (NHG)

[†]These authors contributed equally to this work

**Abstract** Traumatic brain injury (TBI) causes mortality and disability worldwide. It can initiate acute cell death followed by secondary injury induced by microglial activation, oxidative stress, inflammation and autophagy in brain tissue, resulting in cognitive and behavioral deficits. We evaluated a new pomalidomide (Pom) analog, 3,6'-dithioPom (DP), and Pom as immunomodulatory agents to mitigate TBI-induced cell death, neuroinflammation, astrogliosis and behavioral impairments in rats challenged with controlled cortical impact TBI. Both agents significantly reduced the injury contusion volume and degenerating neuron number evaluated histochemically and by MRI at 24 hr and 7 days, with a therapeutic window of 5 hr post-injury. TBI-induced upregulated markers of microglial activation, astrogliosis and the expression of pro-inflammatory cytokines, iNOS, COX-2, and autophagy-associated proteins were suppressed, leading to an amelioration of behavioral deficits with DP providing greater efficacy. Complementary animal and cellular studies demonstrated DP and Pom mediated reductions in markers of neuroinflammation and α-synuclein-induced toxicity.

## Introduction

Traumatic brain injury (TBI) is a principal cause of death and long-term disability in the developed world. Annually, an excess of 10 million people suffer a TBI event worldwide (*Hyder et al., 2007*; *Ruff and Riechers, 2012*), with some 1.6 to 3.8 million cases occurring within the US alone (*Mckee and Daneshvar, 2015*). By far the majority of TBIs are mild to moderate in character and account for 80–95% of cases, with severe TBI comprising the balance (*LoBue et al., 2019*; *Tagliaferri et al., 2006*). With increases in survival rate following initial injury, TBI can result in significant and lifelong cognitive, physical, and behavioral deficiencies that require long-term access to health care and disability services (*LoBue et al., 2019*; *Pavlovic et al., 2019*; *Tagliaferri et al., 2006*). It is now understood that TBI represents a process activated at the time of induction, rather than a single event, and whether clinically manifested or asymptomatic, is one of the most powerful

environmental risk factors associated with the later development of dementia, and particularly Parkinson's and Alzheimer's diseases (*Barnes et al., 2014*; *Gardner and Yaffe, 2015*; *LoBue et al., 2019*), triggered by gene changes in pathways that lead to these disorders (*Tweedie et al., 2013a*; *Tweedie et al., 2013b*).

The incidence of TBI is on the rise (*Coronado et al., 2015*) and by 2020, TBI will comprise the third largest portion of the global disease burden (*Hyder et al., 2007*). With the growing comprehension that a mild or moderate TBI is not benign, and the brain may never fully recover from the injury with time, a greater urgency for an effective treatment has arisen. TBI-associated brain damage is often classified into two key stages. An initial primary damage phase occurs at the moment of insult, comprising contusion and laceration, diffuse axonal injury and intracranial hemorrhage that, combined, induce immediate (necrotic) cell death (*Greig et al., 2014*; *LaPlaca et al., 2007*) that is difficult to reverse. This is followed by a protracted secondary phase that encompasses biological cascades associated with neuroinflammation, ischemia, glutamate excitotoxicity, astrocyte reactivity, and ultimately neuronal dysfunction and apoptosis (*Barkhoudarian et al., 2011*; *Greve and Zink, 2009*; *Morganti-Kossmann et al., 2002*; *Schmidt et al., 2005*). These processes represent targets for potential drug intervention (*Diaz-Arrastia et al., 2014*).

The initiation of an inflammatory response is essential for launching homeostatic neuro-reparative mechanisms following a TBI (*McCoy and Tansey, 2008*; *Mckee and Daneshvar, 2015*; *Schmidt et al., 2005*; *Sullivan et al., 1999*), but if excessive or time-dependently unregulated, these same processes can heighten neuronal dysfunction and degeneration by inducing a self-propagating pathological inflammatory cascade (*Frankola et al., 2011*). Subsequent to a TBI, there is a pronounced elevation in the generation and release of pro-inflammatory cytokines from microglia and astrocytes, and particularly of tumor necrosis factor-α (TNF-α). Brain mRNA and protein levels of TNF-α are markedly elevated shortly after TBI in humans (*Frugier et al., 2010*), and are likewise raised in rodent TBI models. This increase precedes and provokes the generation of multiple other inflammatory cytokines (*Baratz et al., 2015*; *Lu, 2009*; *Shohami et al., 1997*; *Tuttolomondo et al., 2014*). Depending on its concentration, time of release and signaling pathway engaged, TNF-α can, on the one hand, intensify oxidative stress, contribute to glutamate release, and exacerbate blood-brain barrier dysfunction, each of which may promote neuronal dysfunction and loss (*Frankola et al., 2011*; *McCoy and Tansey, 2008*). On the other hand, a delayed, smaller TNF-α rise may augment recovery following TBI, in line with its purported neuroprotective functions, by reducing oxidative stress and promoting neurotrophic factor synthesis (*Patterson and Holahan, 2012*; *Scherbel et al., 1999*; *Sullivan et al., 1999*).

We recently demonstrated that pomalidomide (Pom), an immunomodulatory amino-thalidomide analog that potently inhibits TNF-α synthesis and which is approved for the treatment of multiple myeloma (*Shortt et al., 2013*; *Terpos et al., 2013*), mitigates neuronal loss, neuroinflammation, and behavioral impairments induced by controlled cortical impact (CCI) TBI in rats (*Wang et al., 2016*).

The present study evaluated the actions of a novel analog, 3,6'-dithiopomalidomide (DP), in comparison with Pom (*Figure 1A*) in the same TBI model. We find DP possesses a similar TNF-α lowering action as Pom in RAW 264.7 cellular studies, and in rodents challenged with lipopolysaccharide (LPS), but additionally lowers inflammation-induced COX-2 and iNOS. This broader activity translated to a more potent activity and efficacy, as compared to Pom, in rats challenged with CCI TBI with respect to contusion volume, neuronal cell death and functional outcome evaluated across a broad variety of behavioral, cellular and molecular indices.

## Results

### In vivo studies

### Systemic administration of DP or Pom is well-tolerated and both drugs enter brain

DP and Pom (*Figure 1A*) were well tolerated at the doses evaluated in the present study, 29.5 and 26.4 mg/kg, intraperitoneal (i.p.) injection, respectively: (equimolar to 25 mg/kg thalidomide) in mice and LPS treated rats, and 0.1–0.5 mg/kg intravenous (i.v.) injection in TBI rats, with treated animals being indistinguishable from untreated littermates in relation to a series of assessments focused on health and well-being, as described in *Baratz et al., 2015*. Subjective measures including grooming

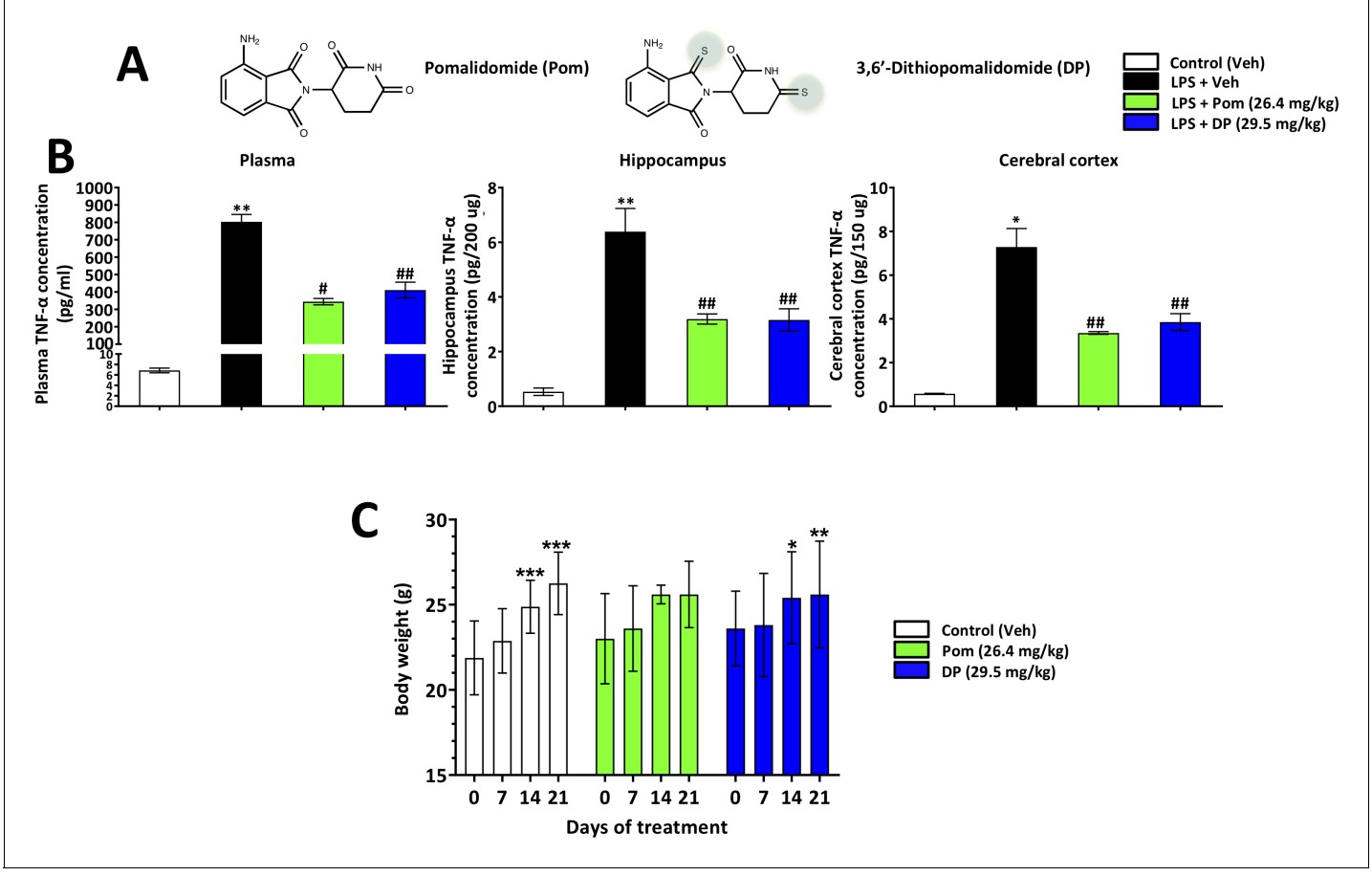

**Figure 1.** Pom and DP, at doses well tolerated by rodents, significantly reduce LPS-induced TNF-α levels in rat plasma, and hippocampal and cerebral cortical tissues. (**A**) Chemical structures of Pomalidomide (Pom) and the novel analog 3,6'-Dithiopomalidomide (DP). (**B**) Treatment of animals with LPS markedly elevated levels of TNF-α. Pretreatment of the animals with Pom and DP significantly reduced these LPS-induced TNF-α increases across plasma and brain tissues. *, ** Refers to the effects of LPS compared to control (Veh). #, ## refers to the effect of drug treatments vs. the LPS + Veh. Values are presented as mean ± S.E.M., of n observations (Veh n = 3; LPS + Veh n = 3; LPS + Pom n = 4; LPS + DP n = 4). (**C**) The effects of daily administrations of Veh, Pom and DP on mouse body weight are illustrated to evaluate drug tolerability. Following 21 days of daily administration there were no adverse effects of agents on animal body weights; animals appeared well groomed and were indistinguishable from one another. *p<0.05, **p<0.01, ***p<0.001 refers to significant changes in body weight compared to body weight at day 0 (i.e. body weight prior to treatment in the same animals). Values are presented as mean ± S.E.M., of n observations (Veh n = 8; Pom n = 5; DP n = 5). There were no significant differences in body weight seen across groups following the same days of treatment.

The online version of this article includes the following source data for figure 1:

**Source data 1.** LPS challenge/drug tolerability.

and appearance, righting skills, ambulation, and blinking reflex, combined with objective measures such as weight (*Figure 1C*).

The brain uptakes of DP and Pom were evaluated in mice administered a systemic dose of 10 mg/kg from plasma and brain samples were obtained at 30 min. DP and Pom were quantified by liquid chromatography with tandem mass spectrometric (LC/MS/MS) detection; DP and Pom possessed a brain/plasma ratio of 0.80 ± 0.08 and 0.81 ± 0.04, respectively; indicating that both compounds similarly readily enter the brain following their systemic administration.

## DP or Pom lower plasma and brain LPS-induced TNF-α levels following their systemic administration

To evaluate the ability of DP and Pom to mitigate elevated systemic and brain levels of TNF-α, each was administered intraperitoneally (i.p.) in conjunction with a dose of LPS (1 mg/kg, i.p.) selected to induce inflammation in rats (*Tweedie et al., 2012*). As illustrated in *Figure 1B*, systemic

administration of either DP or Pom at equimolar doses, 29.5 and 26.4 mg/kg i.p. respectively, significantly lowered LPS-induced TNF-$\alpha$ levels in both plasma and brain. Specifically, LPS elevated plasma, brain hippocampal and cortex levels by 116.5 ± 6, 11.0 ± 1.6 and 12.0 ± 1.5-fold, respectively (for each, p<0.5 vs. control (veh) without LPS). DP inhibited this by 49.3%, 55.6% and 50.9% in plasma, cerebral cortex and hippocampus (p<0.05 vs. LPS alone for each tissue). Pom provided 57.6%, 53.9% and 58.5% inhibition (p<0.05 vs. LPS alone for each tissue); there was no significant difference between DP and Pom, (*Figure 1B* left to right).

## Post-injury administration of DP at 5 hr but not 7 hr mitigated TBI-induced behavioral deficits at 24 hr

To more precisely define a therapeutic time window for treatment, we performed an initial behavioral evaluation of administered DP (0.5 mg/kg, i.v.) vs. Veh-treated rats at 24 hr after TBI with drug administration at either 5 hr or 7 hr following injury. Our DP dose was selected on the basis of our prior study of Pom in the same CCI TBI model (*Wang et al., 2016*). As illustrated by the Elevated Body Swing Test (EBST) in *Figure 2C* (i), an increase in the contralateral swing ratio, indicating body asymmetry, was evident in TBI+Veh rats. DP administered at 5 hr, but not 7 hr, post TBI significantly reduced this ratio from 99.0 ± 1.00% to 66.0 ± 2.92% (p<0.001), with a value of 50% representing that found in normal healthy animals (*Borlongan and Sanberg, 1995*). An impairment in motor coordination, evaluated by measuring beam walking time (*Figure 2C* (ii)), was also observed in TBI + Veh rats. In contrast, TBI + DP animals showed significantly better beam walking performance vs. TBI + Veh treated animals (12.95 ± 0.87 vs. 59.0 ± 1.00 s, p<0.001). Likewise, significant neurological functional deficits, as measured by elevated mNSS scores, were evident in TBI + Veh animals. DP significantly mitigated this deficit with improvement at 5 hr being greater than at 7 hr after TBI (p<0.001 vs. p<0.01; *Figure 2C* (iii)). Finally, a TBI-induced deficit in fine sensorimotor coordination was evident by delayed time for adhesive-removal, as evaluated by a tactile test (*Figure 2C* (iv); 152.4 ± 3.91 vs. 12.6 ± 0.97 s for TBI + Veh vs. sham, respectively, p<0.001). Post treatment with DP at 5 hr, but not 7 hr, likewise mitigated this impairment.

Animals were euthanized after 24 hr post TBI behavioral evaluation and the volume of contusion, quantified from cresyl violet-stained sections, was quantified (ipsilateral as % of contralateral brain region). A TBI-induced loss of cortical tissue was evident in the ipsilateral parietal cortex (*Figure 2A*), which was attenuated by DP at 5 hr (*Figure 2B*; p<0.001) but not at 7 hr. On the basis of these findings, drug administration at 5 hr after TBI was studied in all subsequent experiments.

## Comparative DP and Pom efficacy: Post-injury administration of DP (0.1 mg/kg and 0.5 mg/kg) or Pom (0.5 mg/kg), but not 0.1 mg/kg, significantly reduced TBI-induced contusion volume at 24 hr, with DP thus proving the more effective

We next evaluated TBI-induced contusion volume in the ipsilateral hemisphere at 24 hr after injury following dose-dependent DP or Pom intravenous administration at 5 hr (*Figure 3*). The loss of cortical tissue evident in TBI + Veh animals (19.0 ± 0.74% of the contralateral hemisphere volume (*Figure 3B*)) was significantly mitigated by DP 0.1 and 0.5 mg/kg, reducing the contusion volume to 12.9 ± 0.22% and 7.4 ± 0.87%, respectively. This represents a 32% and 61% reduction compared to TBI+Veh animals (both p<0.001; *Figure 3*). In contrast, Pom 0.1 and 0.5 mg/kg reduced the injury volume to 14.2 ± 0.93% and 11.6 ± 1.02% of the contralateral hemisphere volume, representing a 25% and 39% reduction compared to the TBI + Veh group (p<0.01 and p<0.001; *Figure 3B*). This Pom-mediated protection was thus significantly less than that achieved by the same dose of DP (p<0.05; *Figure 3B*).

Spin-spin relaxation time ($T_2$)-weighted MRI images were obtained from rats in order to cross-validate the CCI TBI-induced injury evident by cresyl violet. As illustrated in the top two images of *Figure 3C*, TBI +Veh-treated animals demonstrated substantial cerebral cortex and hippocampal tissue damage ipsilateral to injury, as evidenced by hyperintensity abnormalities at 24 hr and 7 days following CCI, with the later time showing a greater area of injury. In contrast, a single administration of either dose of DP (0.1 or 0.5 mg/kg) at 5 hr post TBI (*Figure 3C* center and lower panels) resulted in a comparatively smaller injury volume of hyperintensity and associated damage. Sham animals lacked hyperintensities or damage and are not shown.

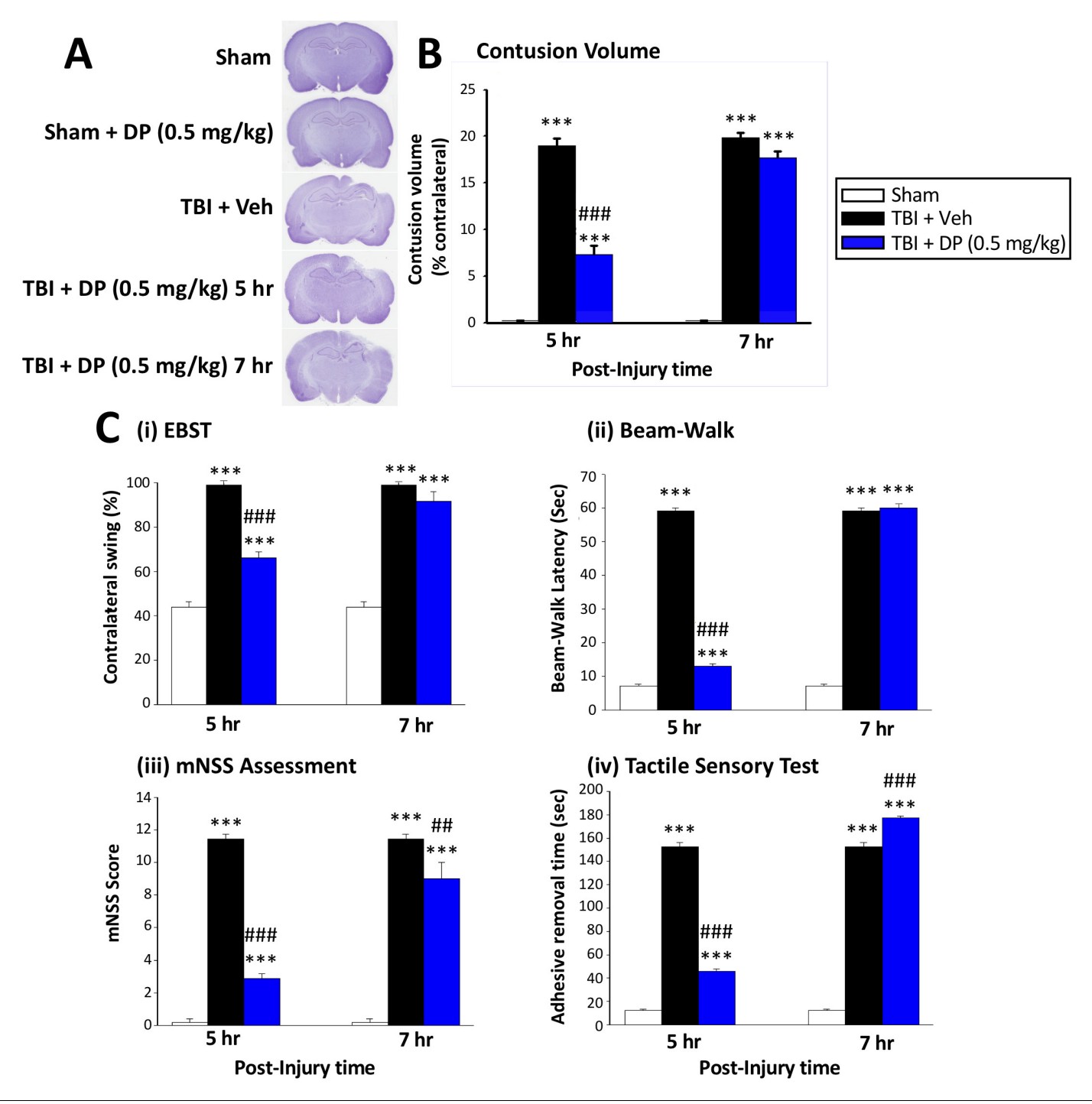

**Figure 2.** Intravenous administration of DP (0.5 mg/kg) at 5 hr but not at 7 hr after TBI significantly reduced contusion volume and behavioral impairments measured at 24 hr after TBI. (**A**) Representative cresyl violet stained coronal brain sections from Sham (control without TBI), DP-treated Sham (Sham + DP 0.5 mg/kg), TBI-vehicle (TBI + Veh), and DP-treated TBI rats (TBI + DP, 0.5 mg/kg) following 5 or 7 hr treatment, (**B**) Intravenous administration of DP (0.5 mg/kg) at 5 hr but not at 7 hr after TBI induced a significant reduction in contusion volume relative to the TBI + Veh group. (**C**) Four separate behavioral outcome measures were quantified to evaluate TBI-induced deficits and their remediation by DP. These involved (i) postural asymmetry, as assessed by the elevated body swing test (EBST), (ii) motor coordination and balance measured by beam walking test, (iii) functional deficits appraised by the mNSS test, and (iv) sensory and motor function, revealed by latency in the adhesive removal test. Across all behavioral outcome measures, DP (0.5 mg/kg) significantly mitigated deficits when administered at 5 hr after TBI. Data represent the mean ± S.E.M. (n = 5 in each group). ***p<0.001 vs. Sham group; ##p<0.01, ###p<0.001 vs. TBI + Veh group.

*Figure 2 continued on next page*

Figure 2 continued

The online version of this article includes the following source data for figure 2:

**Source data 1.** Initial DP TBI activity 5 vs. 7 hr post-insult administration.
**Source data 2.** Initial DP TBI activity 5 vs. 7 hr post-insult administration.
**Source data 3.** Initial DP TBI activity 5 vs. 7 hr post-insult administration.

### Post-injury administration of DP (0.1 mg/kg and 0.5 mg/kg) or Pom (0.5 mg/kg) but not 0.1 mg/kg significantly attenuated TBI-induced behavioral deficits, with DP proving more effective

Illustrated in *Figure 3D*, DP and Pom were without behavioral activity when administered to Sham animals. However, post-treatment with DP (0.1 mg/kg and 0.5 mg/kg) or Pom (0.5 mg/kg) at 5 hr after TBI improved outcomes across all measured behavioral outcomes (neurological function, asymmetrical motor behavior, sensorimotor function and motor coordination (*Figure 3E* (i) – (iv)). Notably, DP (0.1 and 0.5 mg/kg) demonstrated similar or significantly superior efficacy compared to Pom (0.5 mg/kg) across all behavioral tests ($p < 0.01$ and $0.001$), and Pom 0.1 mg/kg lacked significant activity. Consequently, this lower dose of POM was not evaluated further.

### Post-injury administration of DP and Pom attenuated TBI-induced neuronal loss and degeneration

Utilizing an antibody against the neuronal marker NeuN to label neurons, the NeuN-positive (+) cell number was found to be significantly decreased in the cortical contusion margin of TBI + Veh vs. sham animals ($238.20 \pm 11.45$ cells/mm$^2$ vs. $649.74 \pm 43.00$ cells/mm$^2$, $p < 0.001$; *Figure 4*). Treatment with DP (0.5 mg/kg) or Pom (0.5 mg/kg) markedly attenuated this neuronal loss to $523.31 \pm 20.40$ cells/mm$^2$ and $423.93 \pm 46.68$ cells/mm$^2$, respectively. This represents a mitigation of TBI-induced neuronal loss of 69.2% and 45.1%, respectively, vs. the TBI + Veh group ($p < 0.001$ and $p < 0.001$; *Figure 4C*).

We used FJC, a high affinity fluorescent dye for degenerating neurons (*Hall et al., 2008*), to cross-validate the neuronal loss evaluated above by counting NeuN-(+) cells. A quantitative comparison of FJC-(+) cells observed in the cortical contusion area across Sham and TBI groups is shown in *Figure 5*. TBI significantly increased the number of degenerating neurons ($472.9 \pm 48.9$/mm$^2$, $p < 0.001$) compared to the sham group ($15.3 \pm 1.3$/mm$^2$). In contrast, post-treatment of TBI animals with Pom 0.5 mg/kg significantly reduced degenerating neuron number by 52.9% ($231.0 \pm 19.3$/mm$^2$, $p < 0.01$ vs. TBI + Veh), and DP 0.1 and 0.5 mg/kg by 64.9% and 65.6%, respectively ($172.9 \pm 13.2$/mm$^2$ and $176.1 \pm 10.7$/mm$^2$, both $p < 0.001$ vs. TBI + Veh), with again the higher DP dose demonstrating a significantly greater effect than a similar dose of Pom ($p < 0.05$) (*Figure 5C*).

### Post-injury administration of DP or Pom decreased TBI-induced mRNA expression of caspase-3 in ipsilateral cortex, and lowered the fraction of neurons expressing cleaved caspase-3

Caspase-3 is a widely used marker of apoptosis (*Namura et al., 1998*), and hence was employed to further evaluate TBI-induced neuronal loss. Specifically, mRNA levels of caspase-3 were quantified in the cortical contusion margin across animal groups by real-time quantitative RT-PCR at 24 hr after TBI. TBI induced a 2.8-fold elevation in caspase-3 mRNA levels vs. sham ($p < 0.01$) that was significantly reduced by Pom ($p < 0.05$) and fully inhibited by DP ($p < 0.001$; *Figure 6A*). As cleaved caspase-3 is a primary effector of the caspase cascade that leads to apoptosis (*Glushakova et al., 2018*), the co-expression of cleaved caspase-3 and NeuN was evaluated within the cortical contusion margin (*Figure 6B*) to quantify the percent of neuronal cells expressing this activated form of caspase-3. TBI induced a 4.5-fold rise in the fraction of neurons expressing cleaved caspase-3 ($p < 0.001$ vs. sham). Pom (0.5 mg/kg) and DP (0.1 and 0.5 mg/kg) attenuated this increase by 42.3% ($p < 0.05$), 69.6% ($p < 0.01$) and 85.8% ($p < 0.01$), respectively. Notably, again the DP 0.5 mg/kg dose proved to be significantly more efficacious than the similar dose of Pom ($p < 0.05$; *Figure 6C*).

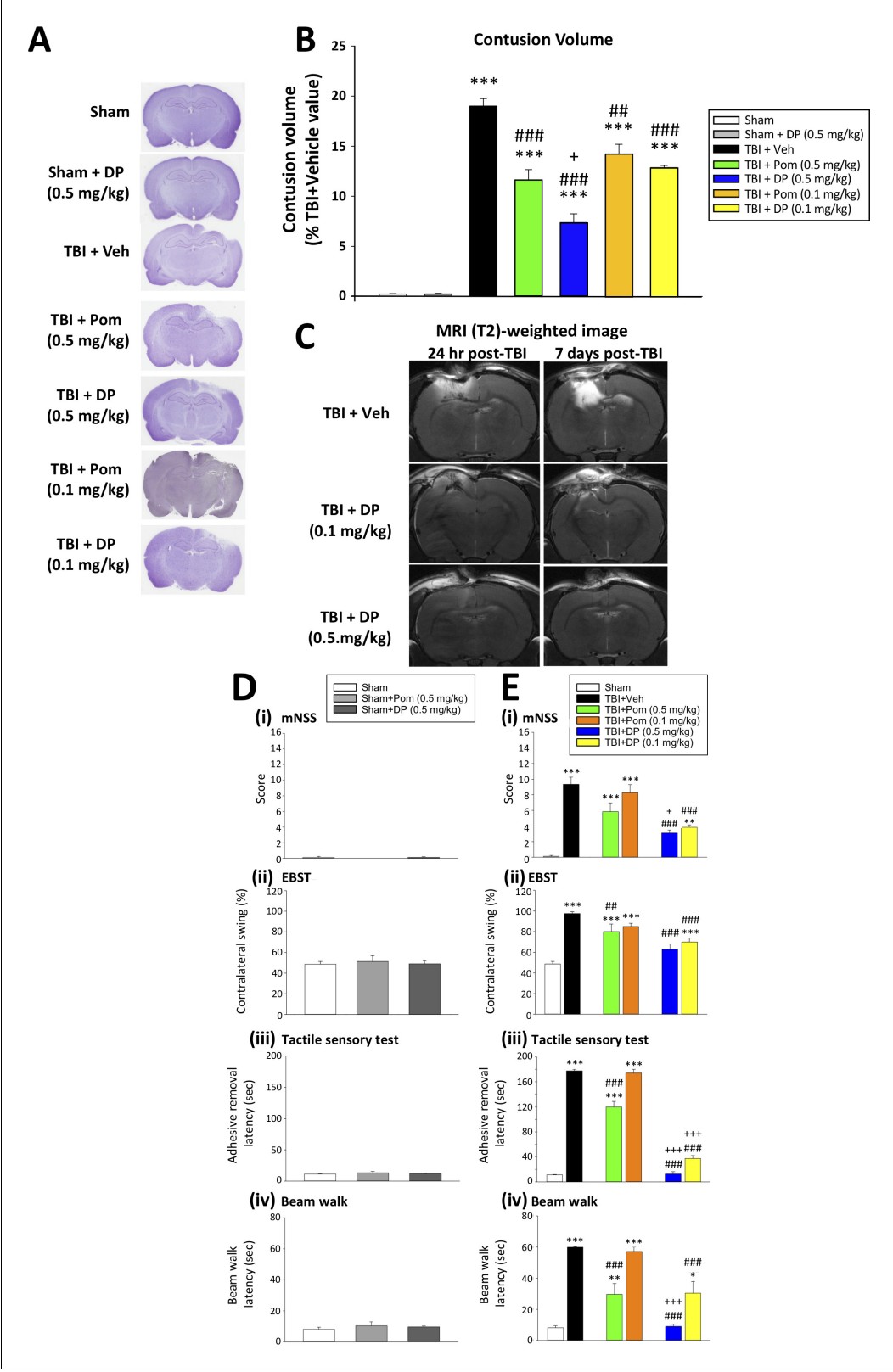

**Figure 3.** DP (0.1 and 0.5 mg/kg) and Pom (0.5 but not 0.1 mg/kg) administered at 5 hr post-injury reduce TBI-induced contusion volume and improve functional outcomes as revealed by behavioral evaluation at 24 hr after TBI. (**A**) Representative cresyl violet stained coronal brain sections from Sham (control without TBI), Sham + DP (0.5 mg/kg), TBI + Veh, and TBI animals treated with DP (0.1 or 0.5 mg/kg) or with Pom (0.1 or 0.5 mg/kg). (**B**) The intravenous administration of DP (0.1 and 0.5 mg/kg) and Pom (0.5 and 0.1 mg/kg) resulted in a significant decrease in contusion volume relative to the

*Figure 3 continued on next page*

*Figure 3 continued*

TBI + Veh group, with DP (0.5 mg/kg) demonstrating significantly greater activity than Pom (0.5 mg/kg). (C) The reduction in contusion volume, evaluated by cresyl violet staining, was cross-validated in a separate cohort of TBI + DP (0.1 and 0.5 mg/kg)-dosed animals by (T$_2$)-weighted MRI imaging at 24 hr and 7 days. Behavioral outcome measures were evaluated across (D) Sham and Sham + DP or Pom (0.5 mg/kg)-treated animals, and in (E) Sham, TBI + Veh, and TBI animals treated with DP (0.1 or 0.5 mg/kg) or Pom (0.1 or 0.5 mg/kg). Notably, DP (0.1 and 0.5 mg/kg) and Pom (0.5 mg/kg) mitigated TBI-induced deficits, with DP (0.5 mg/kg) demonstrating superior activity compared to Pom (0.5 mg/kg); importantly, Pom (0.1 mg/kg) was without efficacy. Behavioral measures involved quantification of (i) mNSS score, (ii) motor asymmetry (EBST), (ii) a tactile adhesive removal test to assess sensory and motor functioning, and (iv) beam walking for motor coordination and balance. Data represent the mean ± S.E.M. (n = 5 in each group). *p<0.05, **p<0.01, ***p<0.001 vs. the Sham group; #p<0.01, ###p<0.001 vs. the TBI + Veh group; +p<0.05, +++p<0.001 vs. the TBI + Pom (0.5 mg/kg) group.

The online version of this article includes the following source data for figure 3:

**Source data 1.** DP vs. Pom (5 hr post-TBI administration) comparable efficacy to mitigate contusion volume & behavioral impairments.
**Source data 2.** DP vs. Pom (5 hr post-TBI administration) comparable efficacy to mitigate contusion volume & behavioral impairments.
**Source data 3.** DP vs. Pom (5 hr post-TBI administration) comparable efficacy to mitigate contusion volume & behavioral impairments.

## Post-injury administration of DP maintained both transcriptional and translational expression levels of p62 as well as reduced LC3-II protein expression in cortex – markers of autophagy

To evaluate the actions of DP and Pom on TBI-induced autophagy, the expression levels of the autophagic markers p62 and LC3-II (*Luo et al., 2011*) were assessed using reverse transcription qPCR (*Figure 7A*) and western blotting (*Figure 7B and C*). TBI-induced autophagy was evidenced by a decrease in the level of p62 (52.0%) in RNA (p<0.001) and 33.7% decline in protein levels (p<0.001)) as well as a 174.0% rise in LC3-II protein levels (p<0.001). DP (0.1 and 0.5 mg/kg)

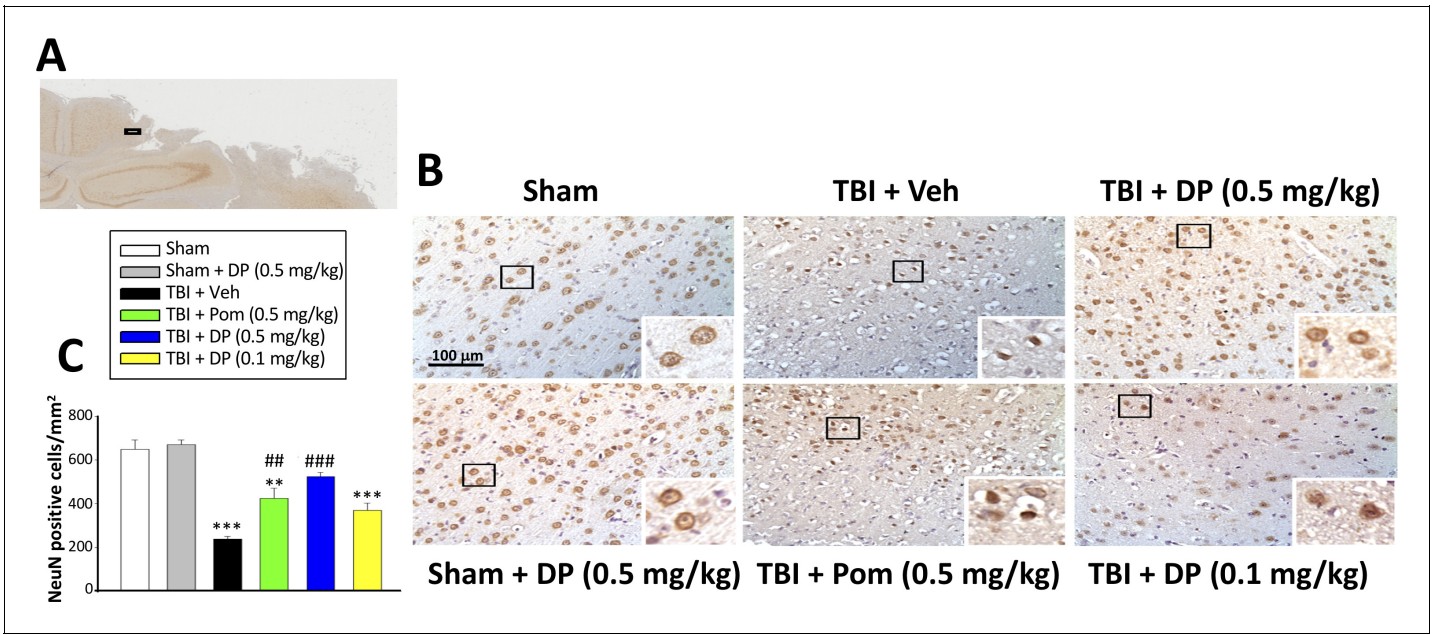

**Figure 4.** Intravenous administration of DP (0.5 mg/kg) and Pom (0.5 mg/kg) at 5 hr after TBI attenuated neuronal loss at 24 hr after TBI. (A) Low power brain section image of NeuN antibody immunohistochemical staining of the cortical region from a TBI animal showing the contusion site and area of observation (Black square). (B) Photomicrographs showing cell structures revealed by NeuN immunocytochemical staining in the ipsilateral cortex. (C) There was a significant decrease in the number of NeuN positive cells in the TBI + Veh group. Five hours post-TBI administration of DP (0.5 mg/kg) or Pom (0.5 mg/kg), with a non-significant trend for DP (0.1 mg/kg), attenuated TBI-induced neuronal loss at 24 hr. Mean ± S.E.M. (n = 4 in each group). **p<0.01, ***p<0.001 vs. the Sham group. ##p<0.01, ###p<0.001 vs. the TBI + Veh group. Scale bar in (B) = 100 μm.

The online version of this article includes the following source data for figure 4:

**Source data 1.** DP vs. Pom (5 hr post-TBI administration) comparable efficacy to mitigate neuronal (NeuN) cell loss.

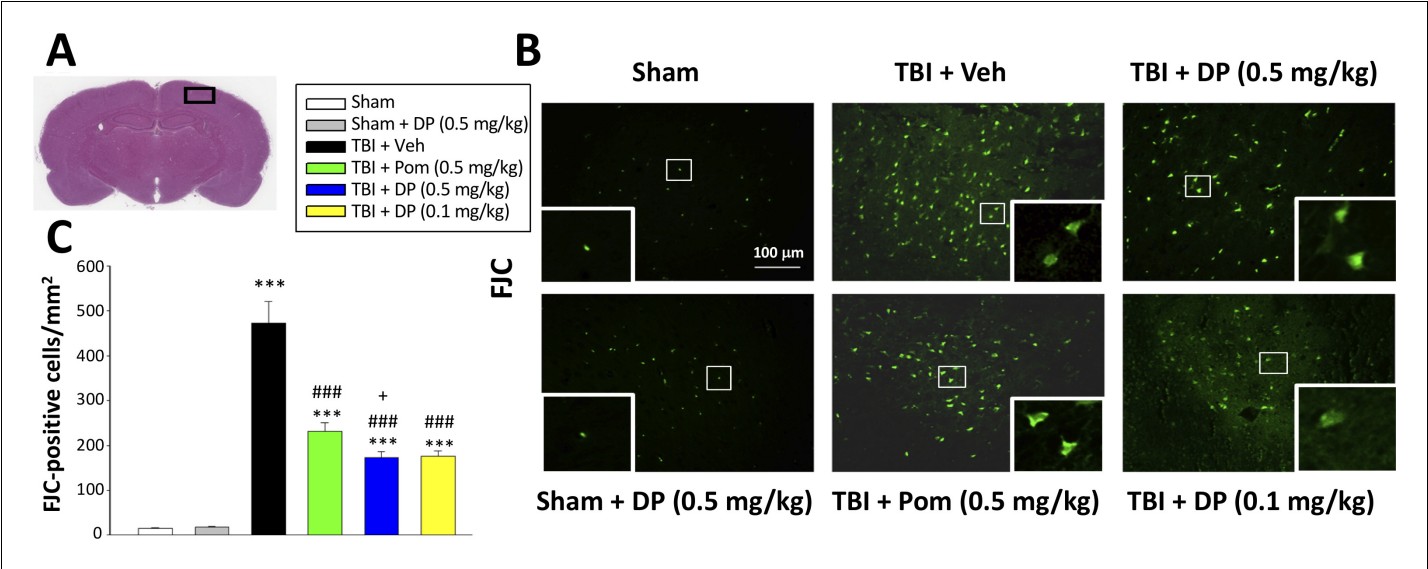

**Figure 5.** FJC staining confirms TBI-induced neurodegeneration within the contusion region, as well as significant mitigation by DP and Pom. (A) Low power HE-stained coronal brain section from a sham animal showing area of evaluation (black rectangular box). (B) Representative photomicrographs showing the FJC-stained cortical region across animal groups at 24 hr after TBI. (C) Quantitative comparisons of mean densities of FJC-positive cells per field in the cortical contusion area at 24 hr post-injury. Notably, DP (0.1 and 0.5 mg/kg) and Pom (0.5 mg/kg) significantly reduced FJC positive neurons within the contusion area, with DP (0.5 mg/kg) showing significantly greater activity than the similar dose of Pom. Mean ± S.E.M. (n = 5 in each group). ***p<0.001 vs. the Sham group. ### p<0.001 vs. the TBI + Veh group. + p<0.05 vs. the TBI + Pom group. Scale bar = 100 μm.
The online version of this article includes the following source data for figure 5:

**Source data 1.** DP vs. Pom (5 hr post-TBI administration) comparable efficacy to lower FJC-labeled degenerating cells.

significantly mitigated these TBI-induced p62 and LC3-II effects (p<0.01 and p<0.05), with Pom (0.5 mg/kg) demonstrating trends that did not reach significance.

## Post-injury administration of DP ameliorated TBI-induced astrogliosis

Astrocytes are key elements in the multicellular responses to CNS trauma, and astrogliosis is a common feature of TBI (*Burda et al., 2016*; *Witcher et al., 2018*). We therefore evaluated GFAP staining to determine TBI and associated drug actions on astrocyte number. TBI induced a 1.8-fold elevation of GFAP expressing cells within the cortical contusion area (p<0.001). Whereas Pom (0.5 mg/kg) had no impact on this TBI-induced increase (p>0.05), DP (0.1 and 0.5 mg/kg) significantly attenuated the elevation in GFAP positive cells by 69.2% and 80.2%, respectively (p<0.001; *Figure 8A and B*). In this regard, both DP doses proved more effective in contrast to Pom (*Figure 8C*).

## Post-injury treatment with DP or Pom mitigated TBI-induced elevations in Iba-1 positive microglial cell number and changed their phenotype from an activated to a resting state

As microglia are essential mediators of the neuroimmune response and are rapidly activated by brain injury (*Frank-Cannon et al., 2009*; *Frankola et al., 2011*; *McCoy and Tansey, 2008*; *Morganti-Kossmann et al., 2002*), we assessed microglial number and morphology by Iba1-immunofluorescence within the margin of the cortical contusion area. As illustrated in *Figure 9B* and quantified in 9E, the vast majority of microglia in sham animals were of a 'resting/quiescent' ramified form composed of long branching processes and a small cellular body. In contrast, microglia within the TBI + vehicle group predominantly demonstrated a rounder cell body and shorter or no processes indicative of the classical 'reactive' amoeboid/round morphology. Notably, and evident in *Figure 9C*, there was a statistically significant 1.95-fold elevation in Iba1-positive cell number in TBI + veh challenged animals (p<0.001). Administration of DP (0.1 and 0.5 mg/kg) or Pom (0.5 mg/kg) significantly attenuated this elevation, inhibiting it by 113.4%, 97.5% and 77.7%, respectively (p<0.001, 0.01 and

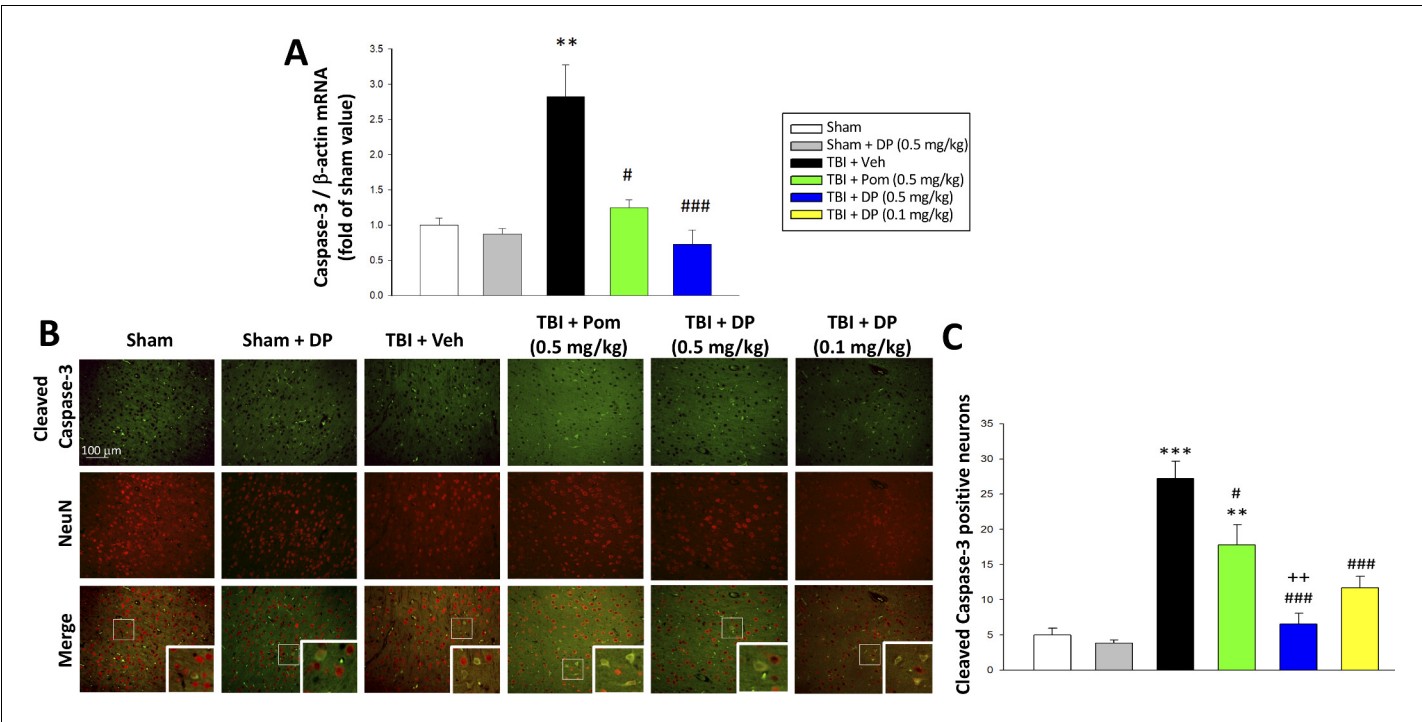

**Figure 6.** TBI induced elevated mRNA expression of caspase-3 and an increase of cleaved caspase-3 protein-containing neurons within the contusion margin. Both were mitigated by DP or Pom. (**A**) TBI induced a marked elevation in mRNA levels of caspase-3, which was fully mitigated by DP (0.5 mg/kg) and significantly lowered by Pom (0.5 mg/kg). Mean ± S.E.M. (n = 5 in each group). \*\*p<0.01 vs. the Sham group. #p<0.05, ###p<0.001 vs. the TBI + Veh group. (**B**) Representative photomicrographs demonstrating, by immunofluorescence, the preferential localization of cleaved caspase-3 protein, a marker of cellular apoptosis, within neurons following TBI. Cleaved caspase-3 immunoreactivity is shown in green, NeuN (a marker for neurons) is shown in red, and yellow indicates colocalization. (**C**) Quantitative comparison of numbers of cleaved-caspase-3 containing neurons per field in the cortical contusion area across animal groups at 24 hr. Notably, DP (particularly the 0.5 mg/kg dose) substantially mitigated the TBI-induced increase in cleaved caspase-3 containing neurons, with greater efficacy than a similar Pom dose. Mean ± S.E.M. (n = 4 in each group). \*\*p<0.01, \*\*\*p<0.001 vs. the Sham group. #p<0.05, ###p<0.001 vs. the TBI + Veh group. ++p<0.01 vs. the TBI + Pom group. Scale bar in (**B**) = 100 μm.

The online version of this article includes the following source data for figure 6:

**Source data 1.** DP vs. Pom (5 hr post-TBI administration) comparable efficacy to reduce cleaved caspase-3 protein-containing neurons.
**Source data 2.** DP vs. Pom (5 hr post-TBI administration) comparable efficacy to reduce cleaved caspase-3 protein-containing neurons.

0.05, respectively) (*Figure 9C*). Also evident was a DP- and Pom-mediated morphological transition of microglia to a less activated state (*Figure 9E*).

## Post-injury administration of DP or Pom significantly attenuated TBI-induced elevations in mRNA expression levels of pro-inflammatory cytokines

As reactive microglia are known to generate elevated levels of pro-inflammatory cytokines and related proteins, we evaluated the mRNA expression levels of TNF-α, interleukin-6 (IL-6) and interleukin-1β (IL-1β) in cortical tissue ipsilateral to brain injury. Each was significantly elevated within the TBI + Veh group, as compared to the sham (TNF-α mRNA: 3.1 ± 0.17 fold, p<0.001; *Figure 10A*); IL-6 mRNA: 3.13 ± 0.44 fold, p<0.001, *Figure 10B*); IL-1β mRNA: 4.43 ± 0.34 fold, p<0.001, *Figure 10C*)). Treatment with DP 0.5 mg/kg, to a lesser extent with DP 0.1 mg/kg, and to a still more reduced amount with Pom 0.5 mg/kg decreased TBI-induced mRNA expression levels of all three of these pro-inflammatory cytokines. In contrast, significant declines in mRNA expression levels of the classical markers of M2-like 'anti-inflammatory' microglia, IL-4 and arginase-1, were evident following TBI (*Figure 10E and F*, p<0.001). These changes, likewise, were significantly mitigated by DP 0.5 (p<0.01 and<0.5, respectively) and less so by DP 0.1 mg/kg and Pom 0.5 mg/kg. Anti-inflammatory IL-10 mRNA levels were largely unaffected by TBI challenge, but were significantly elevated in the TBI + DP 0.5 mg/kg group (p<0.01 and <0.5, respectively, vs. sham and TBI + veh).

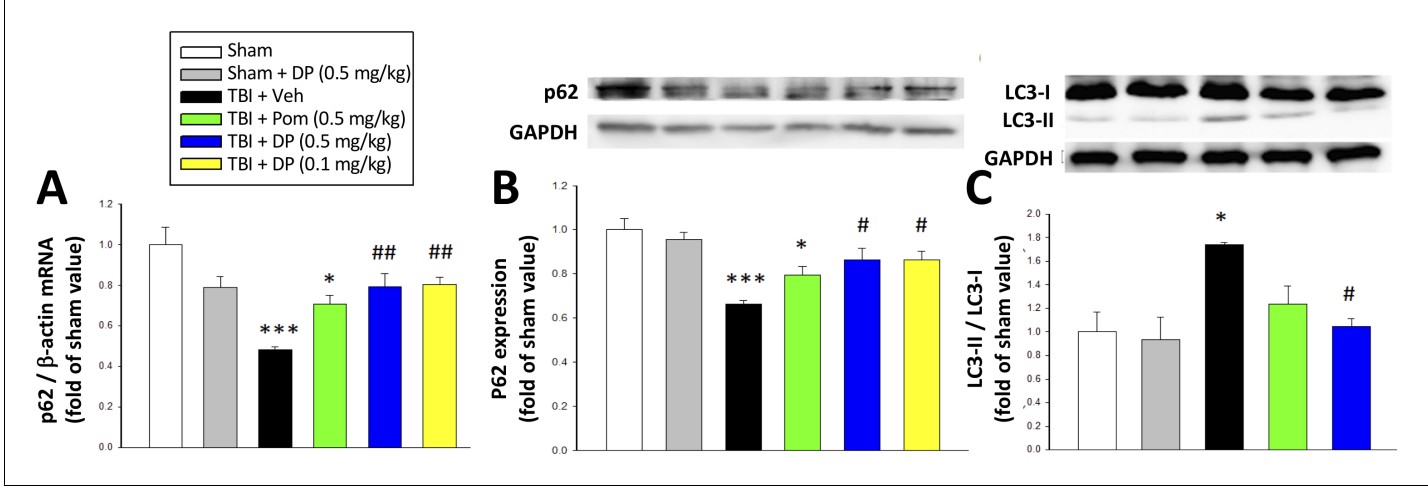

**Figure 7.** Post-injury administration of DP but not Pom attenuated TBI-induced markers of autophagy, as evidenced by mitigation of TBI-induced changes in mRNA and protein expression of p62 and the conversion of LC3-I to LC3-II. (**A**) TBI induced a reduction in the mRNA expression of p62, resulting in (**B**) a decrease in p62 protein levels and, additionally, (**C**) an increase in the expression in the LC3-II/LC3-I ratio (an indicator for late-stage autophagy) in the cortical contusion region at 24 hr. Importantly, each of these TBI-induced changes was significantly inhibited by DP, but not Pom (0.5 mg/kg).Data are expressed as mean ± S.E.M. (n = 5 in each group). *p<0.05, ***p<0.001 vs. the Sham group. #p<0.05, ##p<0.01 vs. the TBI + Veh group.

The online version of this article includes the following source data for figure 7:

**Source data 1.** DP vs. Pom (5 hr post-TBI administration) comparable efficacy to mitigate autophagy impairments.
**Source data 2.** DP vs. Pom (5 hr post-TBI administration) comparable efficacy to mitigate autophagy impairments.
**Source data 3.** DP vs. Pom (5 hr post-TBI administration) comparable efficacy to mitigate autophagy impairments.

As illustrated in *Figure 10G,H and I*, TBI-induced elevations in pro-inflammatory cytokine mRNA expression levels were mirrored in relation to protein levels (p<0.001 for each of TNF-α, IL-6 and IL-1β), were significantly ameliorated by DP 0.5 mg/kg and less so by Pom 0.5 mg/kg.

## Post-injury treatment with DP or Pom significantly mitigated TBI-induced elevations in mRNA expression levels of iNOS, COX2, and of COX2 protein expressing neurons

In the light of prior studies demonstrating that the inflammatory enzymes iNOS and COX2 are upregulated after TBI (*Louin et al., 2006*; *Strauss et al., 2000*) and deleterious in the acute post-traumatic injury phase (*Chen et al., 2007*; *Garry et al., 2015*), we quantified their mRNA expression within the cortical contusion area across all animal groups. As evident in *Figure 11A and B*, TBI induced a significant induction of iNOS (8.5-fold, p<0.001) and COX2 (4.7-fold, p<0.001) mRNA expression. DP 0.5 mg/kg, and to a lesser extent 0.1 mg/kg and Pom 0.5 mg/kg, substantially mitigated this TBI-induced rise. There was inhibition of the iNOS elevation by 92.9% (p<0.001), 27.6% (p<0.01) and 90.6% (p<0.001), respectively; an inhibition of the COX2 elevation by 91.9% (p<0.001), 64.3% (p<0.01) and 56.4% (p<0.001), respectively, was also found.

As post-traumatic neuroinflammation is characterized by COX-2 upregulation (*Strauss et al., 2000*) and the associated generation of thromboxane, prostanoids and free radicals (*Hurley et al., 2002*), we immunohistochemically evaluated COX-2 protein expression in neurons (*Figure 11C*). Whereas some 6.0% of neurons expressed COX-2 in the sham group, this rose to 33.3% subsequent to TBI (*Figure 11D*). DP 0.5 and 0.1 mg/kg and Pom 0.5 mg/kg attenuated this increase, significantly inhibiting it by 80.3% (p<0.001), 73.8%) (p<0.001) and 62.4% (p<0.001), respectively.

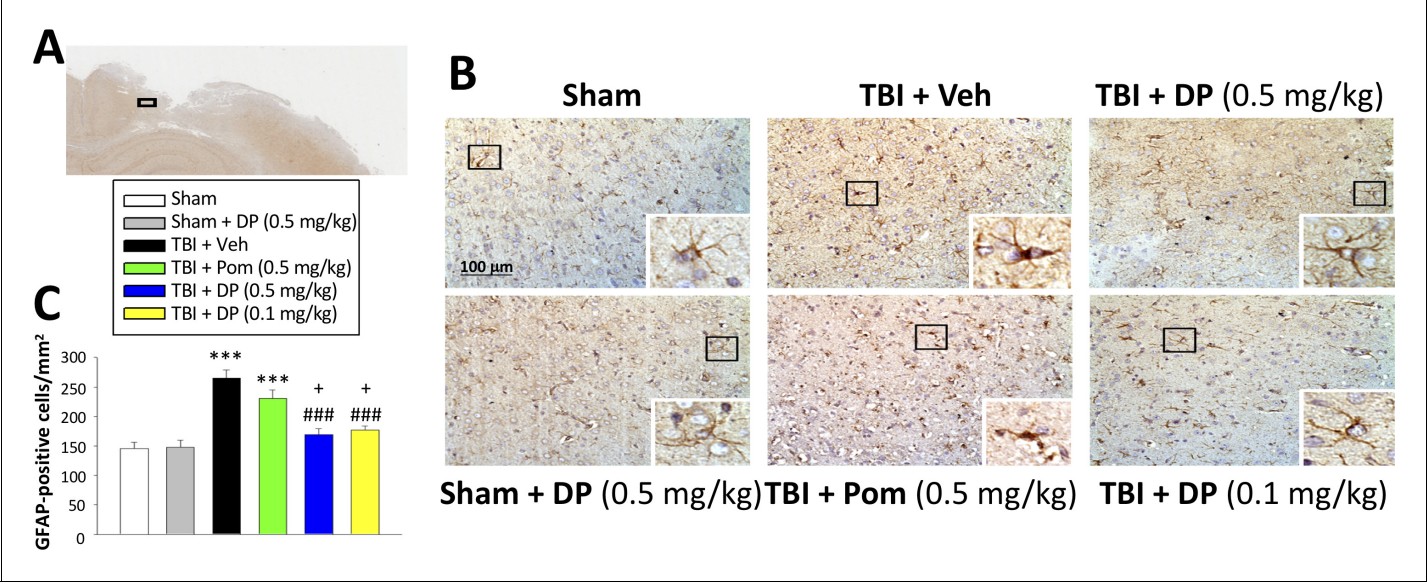

**Figure 8.** Post-injury administration of DP but not Pom ameliorated TBI-induced astrogliosis. (**A**) A low power image of a coronal brain section immunohistochemically stained with GFAP antibody (a marker for astrocytes) in the cortical region from a TBI + Veh animal showing the area of observation (black rectangle). (**B**) Representative photomicrographs showing GFAP immunocytochemical staining across animal groups at 24 hr post TBI or sham challenge, with astrocyte cell morphology magnified in lower right boxes. (**C**) There was a significant elevation in the number of GFAP positive cells in TBI + Veh animals, in which post-injury treatment with DP (0.1 and 0.5 mg/kg) but not Pom (0.5 mg/kg) significantly and substantially mitigated. Mean ± S.E.M. (n = 4 in each group). ***p<0.001 compared with the sham group. ###p<0.001 compared with the TBI + Veh group. +p<0.05 compared with the TBI + Pom group. Scale bar (**B**) = 100 μm.

The online version of this article includes the following source data for figure 8:

**Source data 1.** DP vs. Pom (5 hr post-TBI administration) comparable efficacy to lower astrogliosis (GFAP elevation).

## Cell culture studies

### DP and Pom mitigate markers of inflammation in RAW 264.7 cells challenged with LPS

To further define mechanisms underpinning the neuroprotective actions of DP and Pom, and to differentiate between the two compounds, we evaluated their anti-inflammatory molecular actions in RAW 264.7 cells. This immortal mouse macrophage cell line shares select microglial characteristics and responds to LPS by generating a classical inflammatory response (*Tweedie et al., 2011*). Whereas Pom and DP similarly significantly lowered TNF-α levels induced by LPS challenge (not shown), pretreatment with DP but not Pom induced a substantial and statistically significant 80% decline in nitrite levels which is a stable end product and surrogate measure of nitric oxide (NO) metabolism (*Smith and Lassmann, 2002*), (p<0.001; *Figure 12AB*) This was evaluated further when treatment was initiated 24 hr following LPS challenge to define drug inhibitory actions under conditions where inflammation was fully established, as may occur in TBI. Summarized in *Figure 12C* (Left panel), DP substantially lowered nitrite levels over a 4 hr time period (p<0.05), whereas, in contrast, levels of nitrite in cells treated with Pom continued to rise and were no different from vehicle (DMSO/control) cells. These nitrite lowering actions of DP were cross-validated by means of a different assay for nitrite (*Figure 12D*, Right panel). Illustrated in *Figure 12E*, iNOS levels were elevated in LPS challenged cells and this was substantially inhibited by DP in a concentration-dependent manner. In contrast, Pom was without iNOS lowering actions in LPS-challenged RAW 264.7 cells. Similarly, in *Figure 12F*, COX-2 levels were elevated in LPS challenged cells and substantially inhibited by DP in a concentration-dependent manner. Pom was without COX-2 lowering actions and demonstrated a trend to elevate levels further.

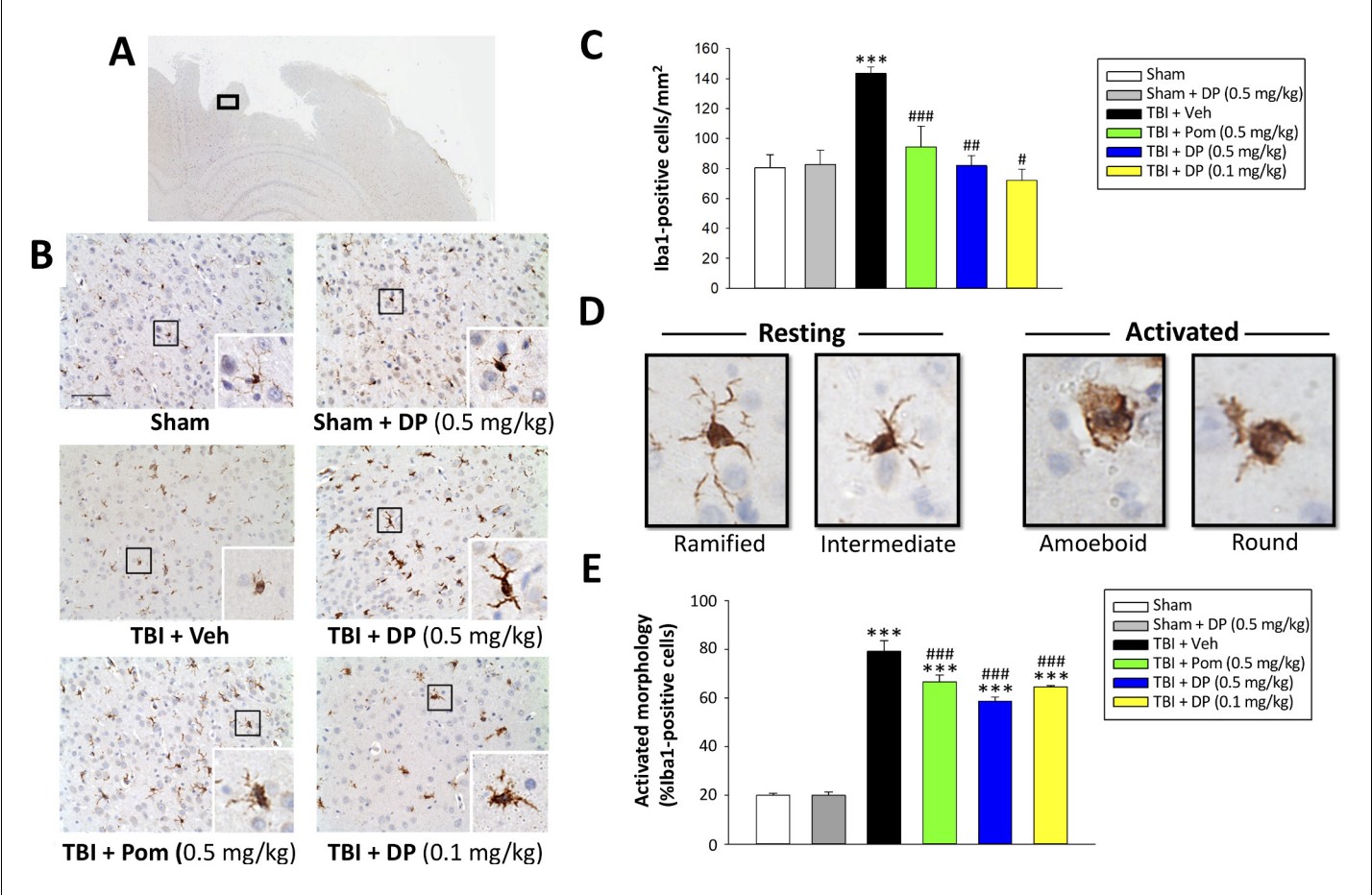

**Figure 9.** Post-injury administration of DP and Pom reduced TBI-induced microglial activation, as evaluated by Iba-1 immunohistochemistry at 24 hr in the cortical contusion area. (A) A low power image of a brain section immuno-histochemically stained with Iba-1 antibody in the cortical region from a TBI + Veh animal showing the contusion site and area of observation (black rectangular box). (B) Representative photomicrographs of immunostaining with Iba-1 showing an increase in microglial activation after TBI, as revealed by an elevation in the number of Iba-1 positive cells/mm² and a change in their morphologic phenotype from a 'resting' ramified/intermediate form in the sham group to a 'reactive' amoeboid/round morphology (detailed in D).(C) Quantification of Iba-1 positive cells/mm² across groups demonstrated a DP (0.1 and 0.5 mg/kg) and Pom (0.5 mg/kg)-mediated reduction in TBI-induced microglial changes. (E) Iba-1-positive cells were morphologically characterized in relation to their activation/resting state and expressed as a percent of their total number. Mean ± S.E.M. (n = 5 in each group). ***p<0.001 vs. the sham group. #p<0.05, ##p<0.01, ###p<0.001 vs. the TBI + veh group. Scale bar (B) = 100 μm.

The online version of this article includes the following source data for figure 9:

**Source data 1.** DP vs. Pom (5 hr post-TBI administration) comparable efficacy to mitigate microglial cell activation.

## DP and Pom mitigate α-synuclein-induced toxicity in primary dopaminergic cultures

An increasing number of epidemiological studies have implicated TBI as a major non-genetic risk factor for later development of neurodegenerative disease (*Delic et al., 2020*), with the strongest evidence associated with PD (*Crane et al., 2016*; *Gardner et al., 2018*). In the light of studies demonstrating the aberrant overexpression of α-synuclein coincident with neuroinflammation in brain following TBI, and suggesting α-synuclein as key a pathological link between TBI and later development of PD-like symptoms in humans and animal models of TBI (*Acosta et al., 2015*; *Wong and Hazrati, 2013*), we evaluated whether DP and Pom could mitigate α-synuclein-induced toxicity in primary neuronal cultures containing dopaminergic neurons. The application of α-synuclein to these primary midbrain cultures induced inflammation and proved toxic for tyrosine hydroxylase (TH)-positive staining neurons, resulting in their decreased survival (*Figure 13A*), a reduction of their neurite network density (*Figure 13B*), and an elevated pro-inflammatory response of OX-41-

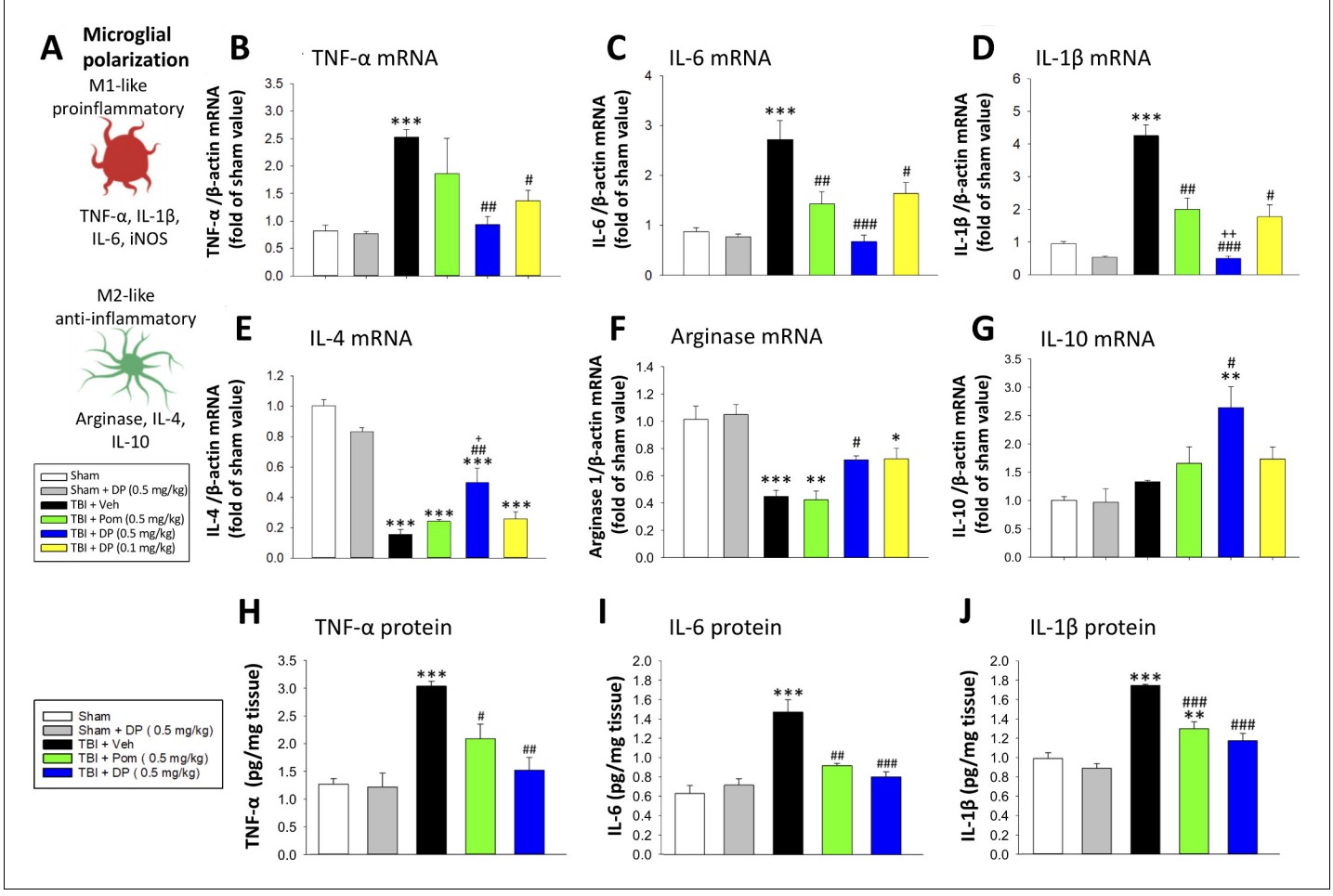

**Figure 10.** Post-injury treatment with DP or Pom significantly reduced TBI-induced cortical pro-inflammatory cytokine mRNA expression and protein levels. (A) The expression levels of markers of a pro- and anti-inflammatory microglial (M1-, M2-like) state were quantified. TBI-induced significant elevations in mRNA levels of (B) TNF-α, (C) IL-6 and (D) IL-1β and declines in (E) IL-4 and (F) arginase-1 within the cortical contusion region, when evaluated 24 hr post injury. These changes were substantially and significantly mitigated by DP (0.1 and 0.5 mg/kg) and, to a lesser degree, by Pom (0.5 mg/kg) treatment, with DP 0.5 mg/kg demonstrating greater efficacy. (G) IL-10 mRNA levels were mildly elevated by TBI, and significantly elevated in the DP (0.5 mg/kg)-treated group. TBI-induced changes in pro-inflammatory cytokine mRNA expression were mirrored in protein levels (H, I, J), and DP (0.5 mg/kg) and, to a lesser extent, Pom (0.5 mg/kg) mitigated these. Mean ± S.E.M. (n = 5 in each group). ***$p<0.001$ vs. the Sham group. #$p<0.05$, ##$p<0.01$, ###$p<0.001$ vs. the TBI + Veh group. ++ $p<0.01$ vs. the TBI + Pom group.
The online version of this article includes the following source data for figure 10:

**Source data 1.** DP vs. Pom (5 hr post-TBI administration) comparable efficacy to normalize brain cytokines.
**Source data 2.** DP vs. Pom (5 hr post-TBI administration) comparable efficacy to normalize brain cytokines.
**Source data 3.** DP vs. Pom (5 hr post-TBI administration) comparable efficacy to normalize brain cytokines.
**Source data 4.** DP vs. Pom (5 hr post-TBI administration) comparable efficacy to normalize brain cytokines.
**Source data 5.** DP vs. Pom (5 hr post-TBI administration) comparable efficacy to normalize brain cytokines.
**Source data 6.** DP vs. Pom (5 hr post-TBI administration) comparable efficacy to normalize brain cytokines.
**Source data 7.** DP vs. Pom (5 hr post-TBI administration) comparable efficacy to normalize brain cytokines.
**Source data 8.** DP vs. Pom (5 hr post-TBI administration) comparable efficacy to normalize brain cytokines.
**Source data 9.** DP vs. Pom (5 hr post-TBI administration) comparable efficacy to normalize brain cytokines.

positive microglial cells (*Figure 13C*). DP and Pom significantly mitigated all three α-synuclein-induced actions in a concentration-dependent manner that, for DP, was optimally achieved at concentrations ranging from 10 to 60 μM. Pom attenuated α-synuclein-mediated toxicities only at doses of 3 and 10 μM, with an inverse U-shaped curve at higher concentrations.

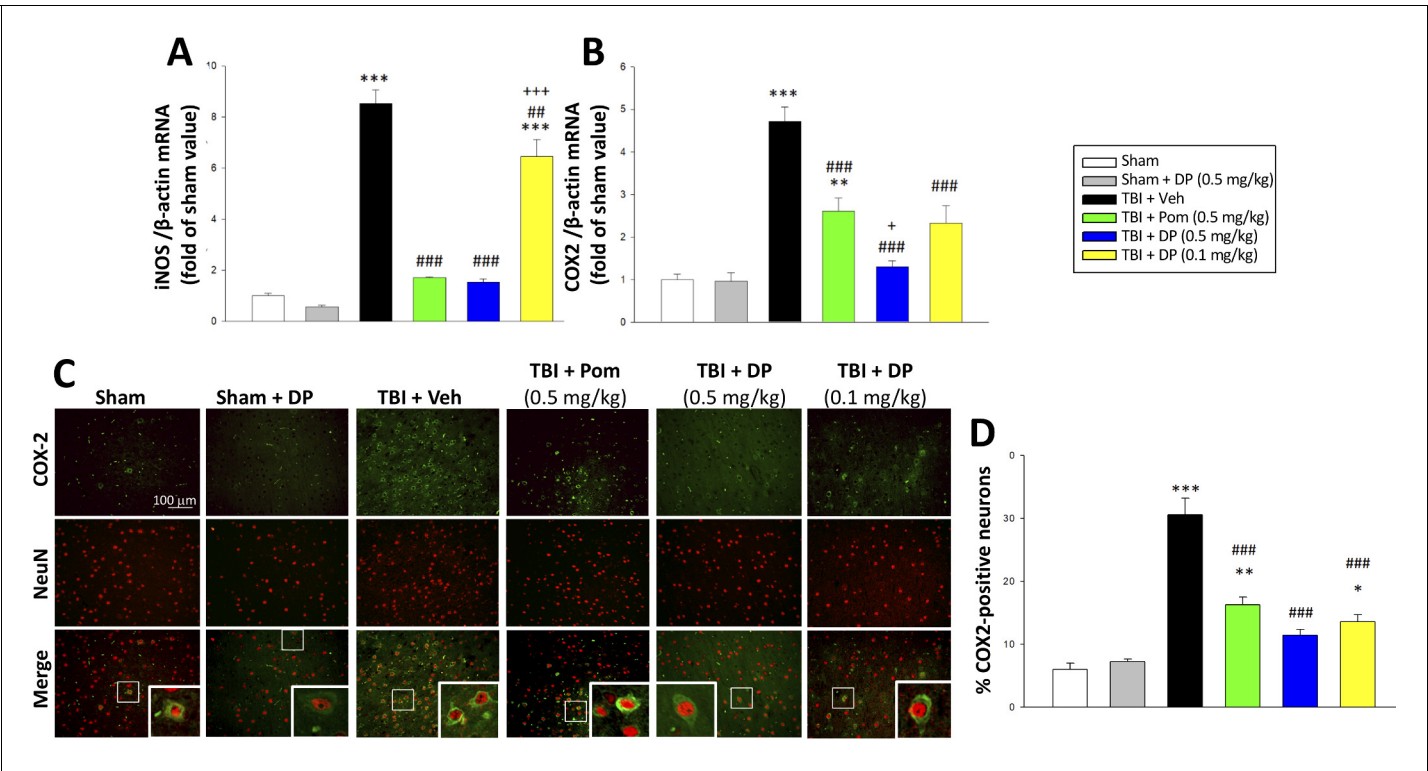

**Figure 11.** TBI-induced elevations in mRNA expression levels of iNOS and COX2, as well as of COX2 protein expressing neurons, which were largely reversed by post-injury treatment with DP or Pom. TBI induced a significant increase in mRNA levels of (A) iNOS, and (B) COX2, markers of inflammation. DP and Pom mitigated these elevations in iNO2 and COX2. Mean ± S.E.M. (n = 5 in each group). **p<0.01, ***p<0.001 vs. the sham group. ##p<0.01, ###p<0.001 compared with the TBI + Veh group. +p<0.05, +++p<0.001 vs. the TBI + Pom group. (C) The immunofluorescence of COX-2 and NeuN protein was evaluated in ipsilateral cortical brain tissue across groups and is shown in representative photomicrographs. COX-2 immunoreactivity is shown in green, and NeuN (a marker for neurons) is shown in red. Colocalization is indicated by yellow. (D) TBI induced a significant increase in the fraction of neurons expressing COX2, which was significantly attenuated by DP and Pom. There was a significant decrease in the number of COX-2 positive neurons in TBI + DP group. Mean ± S.E.M. (n = 4 in each group). *p<0.05, **p<0.01, ***p<0.001 vs. the sham group. ###p<0.001 vs. the TBI + Veh group. Scale bar (B) = 100 μm.

The online version of this article includes the following source data for figure 11:

**Source data 1.** DP vs. Pom (5 hr post-TBI administration) comparable efficacy to normalize brain inflammatory enzymes iNOS and COX2.
**Source data 2.** DP vs. Pom (5 hr post-TBI administration) comparable efficacy to normalize brain inflammatory enzymes iNOS and COX2.
**Source data 3.** DP vs. Pom (5 hr post-TBI administration) comparable efficacy to normalize brain inflammatory enzymes iNOS and COX2.

## Discussion

Accumulating evidence indicates that inflammation contributes to the secondary damage phase that time-dependently occurs following a TBI, and hence neuroinflammation represents a credible therapeutic target for intervention. With more than 2.4 million cases of TBI in the United States annually (*Coronado et al., 2012*), there is a clear and immediate need for an effective TBI treatment strategy. However, despite some 30 prospective, randomized, controlled clinical trials on TBI since 1993, at an estimated cost of $1.1 billion, there are no available approved treatments (*Samadani and Dali, 2016*). Focusing on TNF-α as a drug target involved early in triggering and maintaining neuroinflammation, we synthesized TNF-α synthesis inhibitors aimed to lower the level of its generation but preserve its time-dependent release following an insult to potentially augment the neuro-reparative rather than neurodegenerative features of a neuroinflammatory response following a TBI challenge. The lead compound from these efforts, DP, a novel 3,6' thionated analog of Pom, significantly and dose-dependently reduced contusion volume and improved functional outcomes at 24 hr, with a therapeutic window that closed between 5 and 7 hr in a classical and well characterized rodent model of moderate TBI. Immunohistochemical and biochemical analyses of the cortical contusion

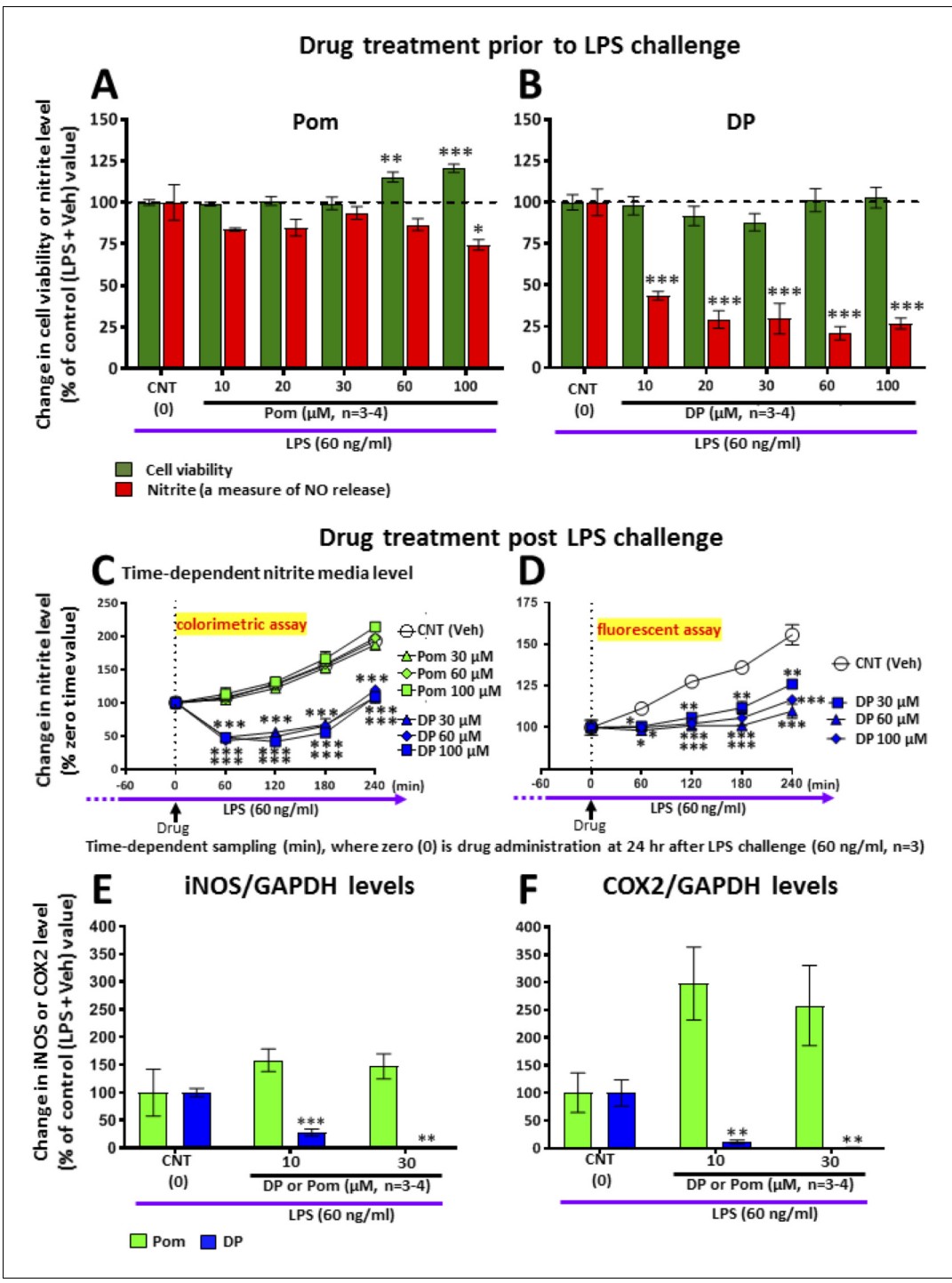

**Figure 12.** LPS challenge in cultured RAW 264.7 cells induced an increase in nitrite (a stable marker of NO), iNOS and COX2, which were mitigated by pre- and post-treatment with DP. Cultured RAW 264.7 cells were pre-treated with either (**A**) Pom or (**B**) DP (10–100 μM) and challenged with LPS (60 ng/ml) 1 hr later. At 24 hr following LPS challenge, cellular viability and nitrite levels, together with (**E**) iNOS and (**F**) COX2 (normalized to GAPDH protein levels) were quantified. DP and Pom were well tolerated at concentrations up to 100 μM, and DP significantly lowered LPS-induced elevations in nitrite, iNOS and COX2 levels. To evaluate whether DP could mitigate LPS-induced nitrite elevation when inflammation was already established, RAW 264.7 cells were challenged with LPS (60 ng/ml) and with post-treatment with DP and Pom 24 hr thereafter (**C, D**). Effects of drugs on nitrite levels were cross-validated using different nitrite assays, DP significantly lowered LPS-induced nitrite levels. *p<0.05, **p<0.01, ***p<0.001 vs. the Control (LPS + Veh) group.

*Figure 12 continued on next page*

*Figure 12 continued*

The online version of this article includes the following source data for figure 12:

**Source data 1.** DP vs. Pom comparable efficacy to mitigate inflammation measures in RAW 264.7 cells challenged with LPS.

tissue demonstrated a DP-mediated amelioration of TBI-induced neuronal cell loss, microglial activation and astrogliosis, and supported the involvement of endogenous compensatory anti-inflammatory, anti-oxidant and autophagy-related cascades, with DP demonstrating greater potency than Pom. The present study hence supports the central role of TNF-α in the neuroinflammatory response that follows a moderate TBI, and introduces a new compound capable of curbing the inflammatory response to mitigate secondary phase injury-associated processes.

Neuroinflammation is a double-edged sword. Post-traumatic inflammation can be advantageous to advance the clearance of cellular debris and promote regenerative processes following a brain insult, such as with TBI. When excessive, however, neuroinflammation can promote neuronal cell death and progressive neurodegeneration (*Simon et al., 2017*). Whereas anti-inflammatory therapies have demonstrated positive results in animal models of TBI, such promise has not translated well into randomized placebo-controlled human clinical trials and, in some cases, may have been detrimental (*Roberts et al., 2004*; *Scott et al., 2018*). The development of the innate immune response following a head injury involves a complex interaction of soluble factors. Cellular lysis occurring during the initial phase of injury, accompanying the primary mechanical insult, allows the release of damage associated molecular patterns (DAMPs). Consistently across rodent TBI models, these trigger the rapid upregulation of TNF-α; both systemic and brain TNF-α levels rise within hours of injury, preceding the rise in other inflammatory cytokines, and ultimately decline to basal levels after approximately 24 hr (*Bachstetter et al., 2013*; *Harrison et al., 2014*; *Rowe et al., 2018*; *Lu, 2009*; *Shohami et al., 1997*). Such a rise in TNF-α can stimulate monocyte infiltration, glial activation, neuronal and myelin loss, and increase BBB permeability (*Chio et al., 2015*; *Rochfort and Cummins, 2015*). TNF-α receptor-mediated NF-KB activation can transcriptionally induce additional TNF-α and thus, amplify TNF-α and receptor signaling pathways to elevate the expression of over 20 different cytokines including IL-1β and IL-6, as evident in *Figure 10*. Such chemokines and receptors drive the immune response which time-dependently underpins the acute, subacute and then chronic neuroinflammation that ensues (*Yamamoto and Gaynor, 2001*). There are many potential targets within the numerous interconnecting cascades, as well as a substantial redundancy between cascades that can, in part, explain the failure of anti-inflammatory strategies to positively impact TBI outcome measures in prior human trials. A not entirely different scenario was formerly faced in the evaluation of anti-inflammatory agents in the treatment of autoimmune disorders such as rheumatoid arthritis prior to successfully targeting TNF-α with anti-TNF-α antibody biologicals (*Feldmann and Maini, 2003*), which now represent one of the most effective and best-selling drug classes worldwide (*Anderson, 2019*).

Pre-clinical studies of the anti-TNF-α biological etanercept support targeting TNF-α as a viable strategy to potentially mitigate TBI across animal models and humans. Its administration to rats following fluid percussion brain injury reduced motor and neurological deficits at 3 days post-injury, consequent to decreased microglia activation and TNF-α generation (*Chio et al., 2013*; *Chio et al., 2010*). More recent studies have cross-validated the efficacy of post-injury etanercept administration across other preclinical TBI models (*Aykanat, 2016*; *Hasturk et al., 2018*; *Perez-Polo et al., 2016*), and open label studies of etanercept in humans with TBI have demonstrated promising results (*Tobinick, 2018*; *Tobinick et al., 2012*) that warrant follow up. These human studies involved perispinal etanercept administration combined with Trendelenburg positioning (a head down tilt) for several minutes to optimize brain delivery of this poorly BBB permeable therapeutic.

An alternative approach to anti-TNF-α biologicals that sequester released TNF-α and, thereby, block its receptor engagement and subsequent signaling pathway, is to lower the rate of synthesis of TNF-α. Thalidomide enhances mRNA degradation of TNF-α post-transcriptionally (*Moreira et al., 1993*; *Rowland et al., 1999*) and, thereby, decreases TNF-α generation and subsequent actions as recently reviewed by *Jung et al., 2019*. Although thalidomide can mediate these actions in cellular studies, translation into animals and humans has proved difficult to achieve to mitigate

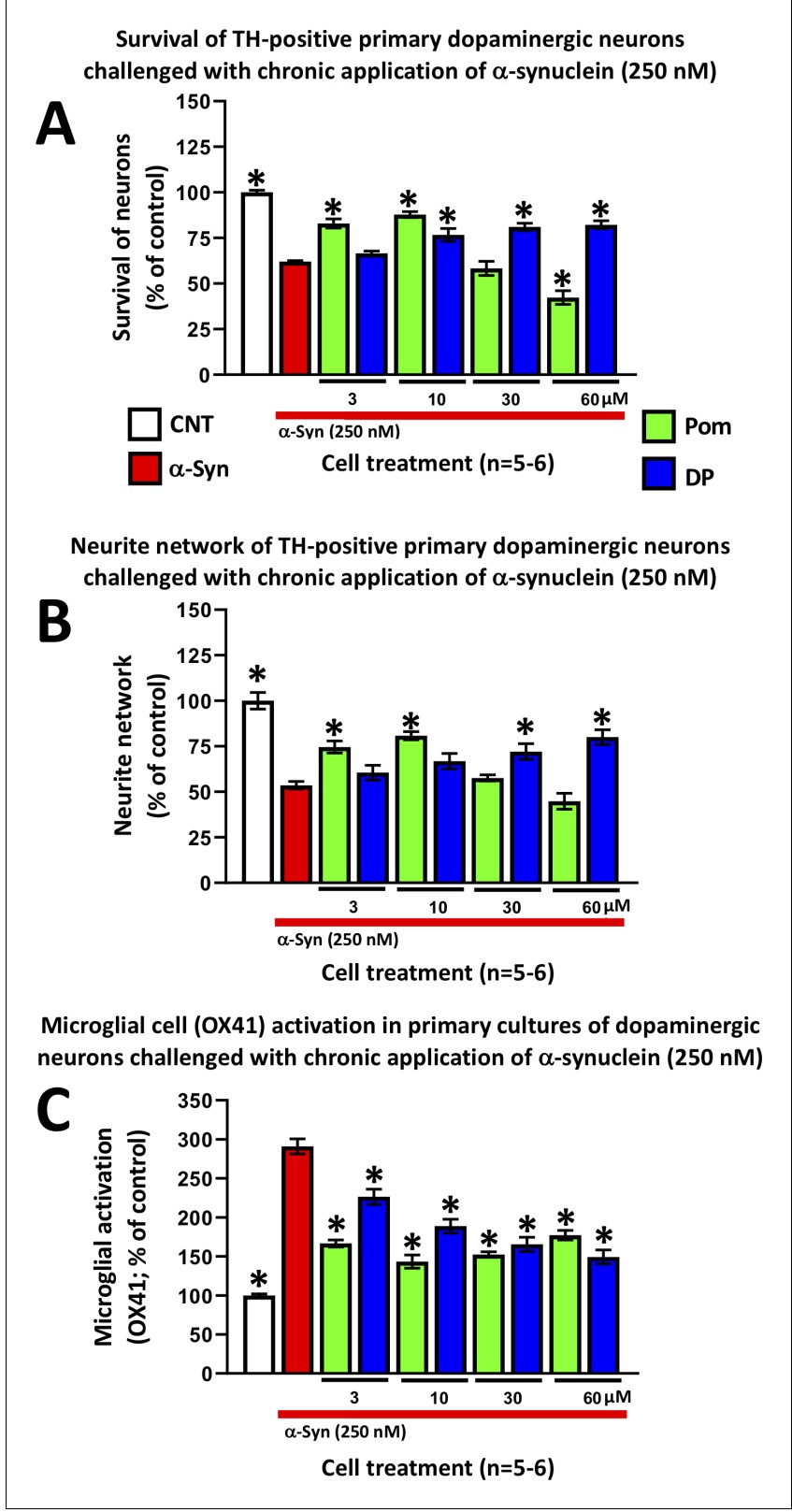

**Figure 13.** DP and Pom mitigate α-synuclein-induced losses of dopaminergic neurons and neurites as well as microglial cell activation in primary cultures. Chronic (72 hr) application of oligomeric α-synuclein (250 nM) to a primary co-culture of dopaminergic neurons and microglia resulted in (**A**) a reduction of dopaminergic neuronal cell survival, (**B**) a reduction in neurite number per cell, and (**C**) an elevated expression of OX41-positive microglial

*Figure 13 continued on next page*

*Figure 13 continued*

cells, indicative of microglial activation. A 1 hr pretreatment with DP or Pom significantly mitigated these α-synuclein-induced effects. Mean ± S.E.M. (n = 5–6 per group). *p<0.05 vs. the α-synuclein alone (red bar) group. The online version of this article includes the following source data for figure 13:

**Source data 1.** DP vs. Pom comparable efficacy to mitigate a-synuclein-induced impairments in primary dopaminergic cell cultures.

neuroinflammation associated with neurodegeneration (*Gabbita et al., 2012*; *Yoon et al., 2013*), with adverse actions proving to be dose-limiting in a recent human AD thalidomide clinical trial (*Decourt et al., 2017*). The thionation of the amide carbonyl groups in key positions (particularly 3,6') within thalidomide dramatically augments TNF-α lowering action (*Tweedie et al., 2012*), as does addition of an amino group to the fourth carbon of the phthaloyl ring, to yield Pom (*Shortt et al., 2013*; *Terpos et al., 2013*). The utilization of both structural changes yields DP (*Figure 1A*).

Like, thalidomide, both DP and Pom appear to readily cross the BBB, with a brain/plasma uptake ratio of approximately 0.8 into normal brain. This brain uptake is in line with the MPO (multiparameter optimization) score for both compounds (5.5 and 4.5, respectively (*Table 1*)). The MPO score provides a composite value of 6 fundamental physicochemical properties that have been found useful to predict whether or not an agent has desirable drug-like properties, and provides a simple and beneficial tool to help prioritize drug candidates for development (*Wager et al., 2016*). A score in the range of 5, which is achieved by both agents, is associated with a favorable likelihood of CNS action. In this regard, when administered as a single systemic dose, both DP and Pom proved capable of lowering both systemic and brain LPS-induced elevations in TNF-α (*Figure 1*) and substantially mitigated secondary phase TBI damage when given 5 hr post injury (*Figures 2–4*). On evaluating dose-dependence, DP demonstrated greater potency than Pom. Whereas DP 0.5 and 0.1 mg/kg doses both significantly reduced TBI-induced contusion volume and behavioral impairments, Pom 0.5 mg/kg but not 0.1 mg/kg proved effective, with Pom (0.5 mg/kg) providing activity similar to the lower DP dose. Furthermore, this dose-dependent Pom activity cross-validates our prior evaluation of Pom in TBI in a separate series of animals (*Wang et al., 2016*). Such cross-validation is critical in the drug development process in the light of the 'reproducibility crisis' impacting the medical sciences (*Stupple et al., 2019*).

Evaluation of a broad number of cellular and biochemical measures was undertaken to define mechanisms underpinning TBI and to further differentiate DP and Pom actions. With regard to drug-mediated reductions in contusion volume, qualitative MRI studies likewise demonstrated DP efficacy at 0.5 and 0.1 mg/kg doses at 24 hr that were maintained at 7 days. Additionally, quantification of NeuN-positive neurons (*Figure 4*) as well as of FJC-positive degenerating cells (*Figure 5*) within the contusion margin, support the differential potencies of DP and Pom.

When cellular damage is sufficiently profound, proapoptotic proteins and transcription factors initiate a process of apoptotic neuronal cell death. The upregulation of caspase-3 in neuronal and glial cells, in particular, contributes to TBI pathology in humans (*Härter et al., 2001*; *Lorente et al., 2015*) as well as in animal models, including CCI (*Glushakova et al., 2018*; *Johnson et al., 2005*; *Newcomb et al., 1999*). In addition to triggering apoptosis, caspase-3 is implicated in the proteolysis of key proteins, such as TDP-43, contributing to TBI and other neurodegenerative disorders

**Table 1.** Six classical physicochemical factors are used to calculate a CNS MPO score that provides a predictive drug-like value.

A set of 6 important physicochemical parameters (i) lipophilicity (cLogP), (ii) calculated distribution coefficient at physiological pH (7.4) (cLogD), molecular weight, (iv) topological polar surface area, (v) number of hydrogen bond donors, and (vi) the most basic center (p$K_a$), are combined together to provide a composite MPO score of predictive drug-like attributes for the central nervous system.

| Compound | cLogP | cLogD (pH7.4) | Molecular weight (g/mol) | Topological polar surface area (Å²) | No. hydrogen bond donors | Strongest basic p$K_a$ calculation | CNS MPO score |
|---|---|---|---|---|---|---|---|
| Pomalidomide | –0.16 | –0.16 | 273.2 | 109.57 | 2 | 1.56 | 4.8 |
| 3,6' Dithiopomalidomide | 0.97 | 0.96 | 305.4 | 5.43 | 2 | 2.33 | 5.5 |

(*Clark et al., 2001*; *Yang et al., 2014*). Our results demonstrate that DP, in particular and more potently than Pom, significantly reduces the number of neurons expressing cleaved caspase-3 (*Figure 6*). Autophagy, an additional process, accelerates the destruction of cells via self-digestion, and is involved in type II (autophagic) cell death, which is distinguishable from type I (apoptotic) cell death (*Shintani and Klionsky, 2004*). We hence quantified both transcriptional and translational expression levels of p62 and protein expression of LC3-II, two classical autophagy markers known to change in human and animal TBI studies (*Wu and Lipinski, 2019*). Declines in p62 and elevations in LC3-II/LC3-I ratio were evident at 24 hr following TBI, and are associated with an enhancement of autophagy initiation in accord with prior CCI TBI studies (*Wu and Lipinski, 2019*; *Zeng et al., 2018*). DP treatment, in particular, normalized levels of both p62 and LC3-II, and thereby lowered TBI-induced autophagy, of potential importance as both excessive as well as too little autophagy appear to contribute to cell death (*Gómez-Díaz and Ikeda, 2019*; *Nopparat et al., 2010*).

An interesting difference between the anti-inflammatory actions of DP versus Pom in our TBI rat model is that, whereas both appeared to equally reduce TBI-induced Iba1 expressing cell number, only DP significantly lowered GFAP expressing cells. Iba1 is a pan-microglial marker whose expression increases with microglial activation (*Hopperton et al., 2018*). TBI induced a change in microglial phenotype to an activated M1-like state (*Figure 9*) in line with the literature (*Madathil et al., 2018*), as well as an elevation in pro-inflammatory mRNA and protein expression (TNF-$\alpha$, IL-1$\beta$, IL-6, *Figure 10*). These changes were largely mitigated by both DP and Pom, with the former consistently demonstrating greater potency. GFAP is a well-characterized marker for mature and differentiated brain astrocytes, and an integral component of their cytoskeleton (*Luoto et al., 2017*). Following brain injury, GFAP is released from degenerating astrocytes, where it can act as a biomarker of damage, and its expression is also elevated in astrocytes that become reactive (*Liddelow and Barres, 2017*; *Luoto et al., 2017*; *Miller, 2018*). Recent studies have suggested that reactive microglia secrete pro-inflammatory proteins that can act together to induce the formation of a subtype of astrocytes (termed A1 astrocytes) that are neurotoxic to neurons and are less capable of supporting neuro-regeneration and promoting new synapses (*Liddelow et al., 2017*; *Witcher et al., 2018*). The activation state of reactive astrocytes is reported to be regulated by NF-$\kappa$B signaling (*Liddelow and Barres, 2017*), and hence the mitigation of astrogliosis by DP is promising.

In the light of the oxidative stress and reactive oxygen species (ROS) induced by TBI (*Khatri et al., 2018*; *Wang et al., 2016*), essential to neuronal function are endogenous anti-oxidant mechanisms and, particularly, cyclooxygenase (COX) expression in the brain, which is associated with inflammatory status (*Figueiredo-Pereira et al., 2015*). COX is generated in two isoforms: COX-1, which largely mediates normal physiological responses, and COX-2, an inducible form that is primarily implicated in pathological conditions. COX-2 is rapidly expressed in several cell types in response to cytokines, growth factors and pro-inflammatory molecules. Its activity can yield ROS and toxic prostaglandin metabolites that, in excess, can exacerbate brain injury (*Hickey et al., 2007*) and drive neurodegenerative processes (*Bartels and Leenders, 2010*). NO similarly, has been associated with the secondary phase of damage after TBI, and is marked by elevated inducible NO synthase (iNOS) in ipsilateral brain regions, (*Üçal et al., 2017*; *Wang and Doré, 2007*); it is generally not detectable in normal brain. Induced by a variety of inflammatory stimuli such as cytokines, iNOS and COX-2 gene expression requires the activation of the transcriptional factor NF-$\kappa$B (*Billiar, 1995*; *Huang et al., 2016*; *Poligone and Baldwin, 2001*; *Taylor et al., 1998*), which, if excessive, can likewise aggravate brain damage after TBI (*Staunton et al., 2018*). Our results demonstrate that DP treatment significantly reduces TBI-induced upregulation of iNOS and COX-2, further supporting that anti-inflammation plays a key role in the mechanisms underpinning the neuroprotective actions of DP. Additionally, COX-2 protein appears to predominantly co-localize with NeuN at 24 hr post TBI, largely within the neuronal cell body, and plays an important role in secondary responses that may result in worsened outcomes (*Dehlaghi Jadid et al., 2019*). Basal levels of COX-2 are found in neurons in cortex and hippocampus, but are normally minimal to undetectable in glia or endothelial cells (*Strauss et al., 2000*). The discrete localization and distinct regulation of COX-2 gene expression in the brain provide a rational basis to suspect that perturbation of prostanoid metabolism may play a significant role in the neuropathology of TBI (*Strauss et al., 2000*). As indicated by our data, the number of COX-2 positive neurons is increased after TBI and this elevation was decreased by DP treatment, indicating that the effects of DP are, in part, associated with actions on COX-2 following TBI.

As TBI-released DAMPs trigger intracellular signaling cascades that lead to microglial activation and the transcriptional induction of downstream pro-inflammatory mediators, particularly COX-2 and iNOS, subsequently causing the subsequent release of cytokines, chemokines, nitric oxide and prostaglandins, we evaluated the comparative actions of DP and Pom on selected aspects of this cascade in cell culture. Both DP and Pom have similar activity in lowering TNF-α levels in cell culture, and do so more potently than thalidomide (*Shortt et al., 2013*; *Terpos et al., 2013*); however in RAW 264.7 cells challenged with LPS, a classical model of inflammation, DP, but not Pom, significantly mitigated elevations in nitrite, iNOS and COX-2. This broader anti-inflammatory action of DP may underpin its greater potency across the majority of the measures evaluated in our in vivo TBI studies.

Beyond the short-term consequences of neuronal dysfunction and loss induced by TBI, biochemical cascades are initiated that can lead to an increased risk of neurodegenerative disorders and, in particular, PD (*Delic et al., 2020*; *Gardner et al., 2018*; *Crane et al., 2016*). α-Synuclein has been proposed as the key pathological link between the chronic effects of TBI and later development of PD (*Acosta et al., 2015*). Elevated α-synuclein expression has been demonstrated in CSF acutely after TBI in infants and children, followed by more chronic elevations (*Su et al., 2010*). A similar overexpression and accumulation of α-synuclein in adults correlated with secondary neuronal cell death, inflammation, development of neuropathological symptoms and reduced survival following severe TBI (*Uryu et al., 2007*; *Mondello et al., 2013*; *Acosta et al., 2015*). TBI-induced elevations in brain α-synuclein levels particularly occur in the cytoplasm of substantia nigra dopaminergic neurons adjacent to the nucleus and along their neurites (*Acosta et al., 2015*; *Wong and Hazrati, 2013*). Elevated α-synuclein levels reduce TH expression levels (*Yu et al., 2004*), alter activity in neuronal networks, disrupt synaptic transmission, and contribute to neuronal cell death (*Hassink et al., 2018*). In the light of these factors, we evaluated the ability of DP and Pom to mitigate the toxicity of aggregated α-synuclein in mixed primary cultures. Both DP and Pom, across a wide concentration range, mitigated α-synuclein-induced microglial activation, thereby cross-validating the anti-inflammatory action of both agents in a second in vitro model. Both agents, additionally, mitigated dopaminergic neuron neurite loss and cell death.

Taken together, our in vivo and cellular in vitro studies focusing on TBI-induced neuroinflammation demonstrate, across a broad number of complementary quantitative behavioral, histological, and biochemical mRNA and protein measures, that Pom and a novel analog, DP, can effectively blunt the neuroinflammatory response. Multiple studies suggest that a neuroinflammatory response is a critical element in the reparative process that follows a brain injury but, when excessive, can exacerbate damage (*Frankola et al., 2011*; *Scherbel et al., 1999*). Pom and, in particular, DP appear to 'recalibrate' the response, moving it within a beneficial restorative range of actions. Our studies with Pom, a clinically approved drug, cross-validate and expand its potential to repurpose as a new treatment strategy for TBI (*Wang et al., 2016*). The first animal studies with DP, reported here, highlight it as an interesting, new and more efficacious analog that warrants further development and evaluation in longer-term follow up studies in TBI as well as in other neurological disorders involving an excessive neuroinflammatory component.

## Conclusion

Our results suggest that post-treatment, in particular, with DP at 5 hr after TBI significantly improves neurological outcome and reduces TBI-induced cell death, neurodegeneration, astrogliosis, microglial activation, and neuroinflammation. Our data reveal that DP and Pom warrant long-term evaluation as a potential therapy to ameliorate TBI-induced functional deficits.

## Materials and methods

POM (4-amino-2-(2,6-dioxopiperidin-3-yl) isoindole-1,3-dione) was synthesized in a two-step process. First, 3-aminopiperidine-2,6-dione was condensed with 3-nitrophthalic anhydride in refluxing acetic acid. Sequential precipitation over ice water (0℃) yielded the resultant nitro-thalidomide as a grey-purple solid. Subsequent hydrogenation over a palladium catalyst provided Pom as a yellow solid, whose structure was confirmed by chemical characterization. DP was then generated from Pom by selective thionation in the 3 and 6' positions, and its structure was similarly confirmed by chemical characterization. The chemical structures of DP and Pom are shown in *Figure 1*.

### In vivo studies

The brain uptake of DP and Pom was evaluated in C57Bl/6 male mice (12 weeks of age, 30 g weight) following administration of 10 mg/kg doses (n = 6) in order to conserve the amount of drug used, as it has been reported that the permeability of the blood–brain barrier is similar between mice and rats, as measured across a wide range of small molecular weight drugs and demonstrating a correlation of 1:1 between the two species (*Murakami et al., 2000*). Primary cell culture and in vivo brain injury studies were performed using Sprague–Dawley rats, as the larger brain of these animals provides a greater abundance of cell or brain tissue for analyses.

All animals were treated according to the International Guidelines for animal research, and the animal use protocol in this study was reviewed and approved by the Institutional Animal Care and Use Committee or Panel (IACUC/IACUP) of both the Intramural Research Program, National Institute on Aging, NIH (protocol No. 331-TGB-2021), and Taipei Medical University [Protocol number: LAC-2015–0051]. Animals were housed in a temperature (21 ~ 25°C) and humidity (45 ~ 50%) controlled room with a 12 hr light/dark cycle and ad libitum access to pellet chow and water. For brain injury we used an animal model of moderate TBI induced by CCI, as described previously (*Huang et al., 2018*; *Yang et al., 2016*; *Yang et al., 2015*) and detailed below.

## Measurement of DP and Pom brain uptake

To evaluate the brain uptake of DP and Pom, plasma and brain drug concentrations were quantified in C57Bl/6 male mice 12 weeks of age, 30 g weight. Administration of 10 mg/kg doses was carried out by the i.p. route as a suspension in 1% carboxymethylcellulose (CMC). This dose was selected based on pilot studies indicating it achieved quantifiable concentrations in plasma and brain. Plasma and samples of cerebral hemispheres were acquired at zero and 30 min (n = 6), immediately placed on wet ice (0° C) and then frozen (−80°C) prior to analysis by LC-MS/MS. In a separate cohort of animals, DP or Pom (29.5 and 26.4 mg/kg, respectively, once daily i.p., n = 5/group) or Veh (l% CMC daily, i.p., n = 8) were administered and mice were evaluated over 21 days to ensure the tolerability of a dose 2.5-fold greater than that selected for the brain uptake study. These doses were chosen as being equimolar to thalidomide 25 mg/kg.

## Sample preparation

To plasma samples of 50 μl, 1.0 ml of ethyl acetate was added and vortexed (10 min). The mixture was then centrifuged at 18,000 × *g* for 10 min, and the organic (upper) phase (850 μl) was separated and placed under vacuum in a centrifugal evaporator to remove solvent. The dried residue was thereafter reconstituted in 60 μl 50% (v/v) acetonitrile in water that contained 50 ng/ml of either butyl nicotinate or ethyl nicotinate (used as internal standards for DP and Pom, respectively). Each sample then was vortexed (4 min), clarified by centrifugation at 18,000 × *g* for 3 min, and subjected to LC-MS/MS analysis.

Weighed, frozen brain samples were sonicated on wet ice in phosphate-buffered saline (PBS: 400 μl per 100 mg brain tissue; 15 s sonication at 4°C). To a 50 μl sample, 1.0 ml of ethyl acetate was then added, vortexed (10 min) and centrifuged at 18,000 × *g* for 10 min. The organic (upper) layer was removed, evaporated to dryness, reconstituted (60 μl 50% (v/v) acetonitrile/water containing 50 ng/ml of appropriate (butyl or ethyl nicotinate)internal standard, centrifuged at 18,000 × *g*, 3 min, and subjected to LC-MS/MS analysis.

## HPLC-MS/MS analysis

HPLC analysis (Waters 2795 Alliance Integrated System) involved separation on a Kinetix C8 (5 μm, 50 × 4.6 mm) column at room temperature by employing a mobile phase gradient of (i) 0.1% (v/v) formic acid in water, and (ii) 0.1% (v/v) formic acid in acetonitrile that was changed from 60%/40% at zero time to 5%/95% at 4 min, and back to 60%/40% at 5.1 min onward with a flow rate 0.3 ml/min. The retention times were, DP: 5.5 min and butyl nicotinate: 5.2 min; Pom: 2.7 min and ethyl nicotinate: 3.1 min. MS compound identification and quantitation were performed on a Micromass Quattro Micro MS by electrospray ionization in the positive ion mode using multiple reaction monitoring (MRM). The MRM transition was for DP: 306.1 to 100.15 m/z (collision energy: 18 eV, cone voltage: 20); butyl nicotinate: 180.1 to 124.1 m/z (collision energy: 25 eV, cone voltage: 25), Pom: 274.1 to 84.15 m/z (collision energy: 13 eV, cone voltage: 20), for ethyl nicotinate: 152.1 to 124.1 m/z

(collision energy: 25 eV, cone voltage: 25). The dwell time was 0.25 s, the capillary voltage 4000, the desolvation temperature 350℃, and the source temperature 115℃. Data quantification used the Quanlynx portion of Masslynx Software version 4.1.

DP and Pom calibration standards (each 0, 10, 50, 100, 250, 500, 1000, 2500, and 5000 ng/ml) were freshly generated using blank mouse plasma or brain samples. These standards were analyzed in duplicate during each analytical run of the collected zero and 30 min time for mouse pharmacokinetic samples of DP and Pom. Each calibration standard curve was prepared by using weighted (1/y) linear regression of the peak area ratio of DP or POM to their appropriate internal standards (butyl and ethyl nicotinate, respectively). No DP or POM signals were apparent in samples generated from zero time plasma or brain samples, or from blank samples not deliberately spiked with either DP or POM. The correlation coefficients for DP and POM were in plasma $r^2$ = 0.998 and 0.994 (10–5000 ng/ml), and brain $r^2$ = 0.998 and 0.995 (50–5000 ng/g), respectively. The lower limit of DP and Pom quantitative detection was 10 ng/ml for plasma and 50 ng/g for brain.

## Systemic and brain LPS anti-inflammatory studies

Male Fischer 344 rats were given DP or Pom (29.6 and 26.4 mg/kg, respectively; equivalent to 25 mg/kg thalidomide, i,p., suspended in 1% CMC) or Veh one hour before the LPS (1 mg/kg, SIGMA, *E. coli* O55:B5). At 4 hr after LPS, animals were euthanized. Plasma was rapidly obtained from heparinized blood; sections of cerebral cortical tissue and hippocampus were dissected on wet ice, and samples were stored at −80℃. Brain samples were sonicated in a TRIS based lysis buffer (Mesoscale Discovery) with 3 x protease/phosphatase inhibitors (Halt Protease and Phosphatase Inhibitor Cocktail, ThermoFisher Scientific), were then centrifuged at 10,000 g, 10 min, 4℃, and protein concentrations were determined using the Bicinchoninic acid assay (BCA, ThermoFisher Scientific). An ELISA for TNF-α was thereafter performed (Mesoscale Discovery) on the rat plasma, hippocampal and cerebral cortical samples, following the manufacturer's protocol. Selection of the LPS and drug doses was based on a former study (*Baratz et al., 2011*).

## TBI studies

Moderate TBI was induced in male Sprague–Dawley rats (250–300 g, body weight) that were anesthetized with chloral hydrate (400 mg/kg; Sigma, St. Louis, MO) and placed in a stereotaxic frame. Specifically, a 5 mm craniotomy was performed over the left parietal cortex, centered on the coronal suture, and 3.5 mm lateral to the sagittal suture. TBI was induced by use of a CCI instrument with a rounded (5 mm diameter) metal tip propelled at a velocity of 4 m/s to a depth of 2 mm below the dura, as described previously (*Huang et al., 2018*; *Yang et al., 2016*; *Yang et al., 2015*). Sham animals received anesthesia and craniotomy but no TBI. To control for fluctuations in animal body temperature consequent to anesthesia or TBI or to drug interactions, rectal temperature was monitored and maintained at 37.0 ± 0.5℃ by use of a heated pad. Subsequently, animals were placed in a heated cage to maintain body temperature while recovering from anesthesia. To further assess for potential changes in body temperature induced by DP, Pom and anesthesia, a series of sham and TBI animals were given both doses of drug and their body temperature was, likewise, monitored over a 3 hr interval. There were no changes in body temperature induced by DP or Pom. Following sham or TBI procedures, animals were randomly assigned to treatment or veh (control) groups (n = 5 per group, based on both prior studies [*Huang et al., 2018*; *Yang et al., 2016*; *Yang et al., 2015*]) and a power analysis (*Charan and Kantharia, 2013*). Behavioral evaluations were undertaken prior and 24 hr following TBI and, thereafter, animals were euthanized for further histological and biological analyses. DP and Pom were administered to a separate cohort of rats that were evaluated for tolerability over the following week (n = 4).

## Behavioral evaluation of neurological outcomes

Behavioral testing was performed before TBI and at 24 hr after TBI. The evaluations consisted of an elevated body swing test (EBST), a tactile adhesive removal test, a beam walk test and a modified neurological severity score (mNSS) assessment. All were performed by an observer blinded to the experimental groups. These procedures were performed as previously described with some modifications (*Li et al., 2018*; *Wang et al., 2016*; *Yang et al., 2015*).

## Contusion Volume

To measure the volume of TBI-induced injury in the ipsilateral cortex 24 hr after TBI, cresyl violet-stained sections were digitized and analyzed using Image J (National Institutes of Health, Bethesda, MD). The volume was computed by adding the injury areas multiplied by the inter-slice distance (500 µm). Hemispheric tissue loss was expressed as a percentage that was calculated by the following formula: [(contralateral hemispheric volume−ipsilateral hemispheric volume)/(contralateral hemispheric volume)×100%], as previously reported (*Zhang et al., 1998*). We used this protocol as we described previously (*Huang et al., 2018*; *Wang et al., 2016*; *Yang et al., 2015*).

## Brain magnetic resonance imaging (MRI) analysis

Animals subjected to TBI and treated with DP (0.1 or 0.5 mg/kg at 5 hr) were evaluated by MRI at 24 hr and 7 days after TBI. Each rat was anesthetized with 2.5% isoflurane by mechanical ventilation and placed in a dedicated holder that was then positioned at the isocenter of a 7.0 Tesla small animal MRI scanner (70/16 PharmaScan, Bruker Biospin GmbH, Germany) with a 72 mm volume coil as the transmitter, and a rat surface coil as the receiver. Scans were obtained from the brain stem to the olfactory bulb with fast spin-echo T2-weighted imaging pulse sequences (TR/TE, 3000/37 ms). Multi-slice images were acquired with a field of view of 20 × 20 mm and with a slice thickness of 1 mm with no gap. The pixel matrix was 256 × 256. These Bruker-obtained images were then converted to DICOM format by using the software program (Paravision 6.0.1) included with the scanner.

## Fluoro-Jade C (FJC) staining

FJC, a derivative of polyanionic fluorescein, selectively binds to degenerating neurons. Using a FJC ready-to-dilute staining kit (TR-100-FJ, Biosensis), we identified degenerating neuronal cells in cortical tissue according to the manufacturer's protocol with some modifications (*Wang et al., 2016*; *Yang et al., 2015*). Brain sections from the different treatment groups were deparaffinized, rehydrated, and incubated in distilled water for 3 min, and then were incubated in a solution of potassium permanganate (1:15) for 10 min. They were next rinsed in distilled water for 2 min, and then incubated in the FJC solution (1:25) for 15 min. After incubation, slides were washed and mounted on coverslips with Vecta-shield mounting medium (Vector Laboratories, Burlingame, CA, USA). All sections were observed and photographed using a fluorescent inverted microscope (IX70, Olympus, Japan). Numbers of FJC-positive cells were counted in three randomly selected fields per slide with SPOT image analysis software (Diagnostic Instruments). Numbers of FJC-positive cells observed on the slides from different blinded treatment groups were counted and used to generate a mean number per treatment group.

## RNA Extraction, Reverse Transcription, and Real-Time Quantitative PCR (qPCR)

Total RNA was extracted by using TRIzol reagent (Life Technologies, Carlsbad, CA, USA). The purity and quality of RNA were confirmed by defining the ratio of absorbance at 260 and 280 nm wavelengths (NanoDrop ND-1000, Thermo Scientific). A sample of 3 µg total RNA was reverse transcribed into cDNA. For mRNA measurement, diluted cDNA was amplified using the Rotor-Gene SYBR Green PCR Kit (Qiagen) in a Rotor-Gene Q 2plex HRM Platform (Qiagen). Reaction conditions were carried out for 35–40 cycles (5 min at 95℃, 5 s at 95℃ and 10 s at 60℃). All procedures for RNA extraction and qPCR have been described previously (*Wang et al., 2016*; *Yang et al., 2016*; *Yang et al., 2015*).

The primers were designed using previously reported cDNA sequences:

1.Caspase-3
Forward - 5' AAT TCA AGG GAC GGG TCA TG 3'
Reverse - 5' GCT TGT GCG CGT ACA GTT TC 3'
2.p62
Forward - 5' AAG TTC CAG CAC AGG CAC AG 3'
Reverse - 5' AGC AGT TAT CCG ACT CCA TCA G 3'
3.β-actin
Forward - 5' GAC CCA GAT CAT GTT TGA GAC CTT C 3'
Reverse - 5' GAG TCC ATC ACA ATG CCW GTG G 3'

4.TNF-α,
Forward - 5' CTC TTC TCA TTC CCG CTC GTG 3'
Reverse - 5' GGA ACT TCT CCT CCT TGT TGG G 3'
5.IL-6
Forward - 5' TTC TTG GGA CTG ATG TTG TTG AC 3'
Reverse - 5' AAT TAA GCC TCC GAC TTG TGA AG 3'
6.IL-1β
Forward - 5' GTT TGA GTC TGC ACA GTT CCC 3'
Reverse - 5' CAA CTA TGT CCC GAC CAT TGC 3'
7.iNOS
Forward - 5' TCT GTG CCT TTG CTC ATG ACA 3'
Reverse - 5' TGC TTC GAA CAT CGA ACG TC 3'
8.COX2
Forward - 5' CGG AGG AGA AGT GGG GTT TAG GAT 3'
Reverse - 5' TGG GAG GCA CTT GCG TTG ATG G 3'
9.IL-4
Forward - 5' GCA ACA AGG AAC ACC ACG GAG AAC 3'
Reverse - 5' CTT CAA GCA CGG AGG TAC ATC ACG 3'
10.Arginase-1
Forward - 5'CAT ATC TGC CAA GGA CAT CG 3'
Reverse - 5'GGT CTC TTC CAT CAC TTT GC 3'
11.IL-10
Forward - 5'ACT GCT ATG TTG CCT GCT CTT 3'
Reverse: 5'ATG TGG GTC TGG CTG ACT GG 3'

## Immunohistochemistry (IHC) and Immunofluorescence (IF)

Brain sections from sham, TBI rats, and TBI rats treated with either DP or Pom were deparaffinized and rehydrated. Serial sections (10 μm) through the cerebral cortex were stained with hematoxylin and eosin for microscopic evaluation. We used immunohistochemistry (IHC) or immunofluorescence (IF) to detect the protein expression of Iba-1, NeuN, GFAP and COX-2 in the ipsilateral lesioned cortex of rats. For immunohistochemistry (IHC), we used rabbit polyclonal anti-Iba1 (GeneTex, GTX100064, 1:200), rabbit polyclonal anti-GFAP (GeneTex, GTX108711, 1:400), and mouse monoclonal anti-NeuN (GeneTex, GTX30773, 1:400) as the primary antibodies. After incubation at 4℃ overnight, the sections were then washed and incubated with VECTASTAIN Elite ABC HRP Kit (Peroxidase, Rabbit IgG and Mouse IgG) at room temperature for 1 hr.

For immunofluorescence (IF), we used the following primary antibodies: (i) rabbit polyclonal anti-COX2 (Abcam, ab15191, 1:400), (ii) mouse monoclonal anti-NeuN (GeneTex, GTX30773, 1:400) for the primary antibodies. After incubation at 4℃ overnight, the sections were then washed and incubated with: (i) Alexa Fluor 488 goat anti-rabbit IgG (Jackson, 1:200), (ii) Alexa Fluor 594 goat anti-mouse IgG (Jackson, 1:200) at room temperature for 1 hr. We then counted the sections using SPOT image analysis software (Diagnostic Instruments, Sterling Heights, MI) as described previously (*Wang et al., 2016*; *Yang et al., 2016*; *Yang et al., 2015*). Controls consisted of omitting the primary antibodies and observers were blinded as to the treatment groups.

## Western Blot analysis

Total proteins were extracted using protein extraction buffer (Mammalian Cell-PE LBTM, Geno Technology, USA) containing protease and phosphatase inhibitors (Complete Mini, Roche Diagnostics, Indianapolis, IN, USA). Proteins were separated by gel electrophoresis, and then electroblotted onto polyvinylidene difluoride (PVDF) membranes (PerkinElmer Life Sciences, USA). In this study, we used the following primary antibodies: (i) Rabbit Polyclonal LC3 Antibody (Cell Signaling, #2775, 1:1000), (ii) Mouse Monoclonal Anti-SQSTM1/p62 antibody (Abcam, ab56416, 1:1000), (iii) Rabbit Polyclonal GAPDH antibody (GeneTex, GTX100118, 1:10000). After incubation at 4℃ overnight, we used the following HRP conjugated secondary antibodies: (i) Goat Polycolonal anti-rabbit IgG (GeneTex, 1:5000) for LC3 and GAPDH, (ii) Goat Polycolonal anti-mouse IgG (GeneTex, 1:5000) for p62. We used GAPDH, a protein that generally expresses in all eukaryotic cells, as an internal control. All bar graphs for western blots in this paper were thus compared with the expression of GAPDH.

## In vitro studies

### DP and Pom activity in LPS stimulated RAW 264.7 cells

Mouse RAW 264.7 cells were obtained from ATCC (Manassas, VA, USA (ATCC TIB-71 certificate of analysis: https://www.atcc.org/Products/All/TIB-71.aspx#documentation; these cells tested negative for mycoplasma contamination as evaluated by both Hoechst DNA stain (indirect) Agar culture (direct) techniques). They were grown in DMEM medium supplemented with 10% FCS, penicillin 100 U/ml and streptomycin 100 µg/ml (to ensure their maintenance under an uncontaminated condition), and were maintained at 37°C and 5% $CO_2$. Cells were propagated in accord with ATCC guidelines, and were cultured as described previously (*Tweedie et al., 2011*). RAW 264.7 cells were challenged with LPS (Sigma, St Louis, MO: serotype 055:B5) at a final concentration of 60 ng/mL. This concentration of LPS in RAW 264.7 cells induces a sub-maximal rise in both TNF-α and nitrite levels without a loss of cell viability. Such a sub-maximal rise is useful for assessing whether the addition of an experimental drug can either lower or further raise levels of TNF-α and nitrite. In the drug 'pre-treatment' study, either DT or Pom (10–100 µM) or Veh, n = 3–4, was administered 1 hr prior to LPS challenge (sample size was selected on the basis of our prior studies [*Tweedie et al., 2011*; *Tweedie et al., 2012*] and a power analysis [*Charan and Kantharia, 2013*]). At 24 hr following the addition of LPS, conditioned medium was harvested and analyzed for quantification of secreted TNF-α protein, nitrite levels, iNOS levels, and COX-2 that were normalized to a housekeeping protein (GAPDH) whose level was found unchanged by either DP, Pom, or LPS challenge. Nitrite ion detection in drug treated RAW 267.4 cell culture medium was assessed by two different assays (1) a colorimetric assay (Griess Reagent System, Promega, Madison, WI, Cat # G2930) and (2) a fluorescence assay (Nitrate/Nitrite Fluorometric Assay Kit, Abnova, Cat # KA1344). The protocols followed were those recommended by the manufacturers. In brief a $NO_2^-$ ion standard curve was prepared and the unknown medium sample $NO_2^-$ levels were determined for the standard curves. Fresh medium was replaced into the wells and cell viability was then assessed by a CellTiter 96 AQueous One Solution Cell Proliferation Assay (Promega, Madison, WI). In a drug 'post-treatment' study, either DP or Pom (30–100 µM) or Veh, n = 3, was administered 24 hr following a LPS challenge, and medium was time-dependently sampled thereafter and quantified for nitrite levels as described above.

### DP and Pom activity in primary dopaminergic neurons and microglia challenged with α-synuclein

Rat dopaminergic neurons and microglia were cultured as described by *Zhang et al., 2005* and *Callizot et al., 2019* with modifications. Fetuses (14 days gestation), aseptically removed from euthanized, timed pregnant female rats, were immediately placed in ice-cold L15 Leibovitz medium containing 2% penicillin (10,000 U/mL), streptomycin (10 mg/mL) and 1% bovine serum albumin (BSA). The ventral portion of the mesencephalic flexure, a midbrain region abundant in dopaminergic neurons, was used as a source for cell preparation and dissected portions were treated for 20 min at 37°C with a trypsin/EDTA solution (0.05% trypsin and 0.02% EDTA). Dulbecco's modified Eagle's medium (DMEM) containing DNAase I grade II (0.5 mg/mL) and 10% of fetal calf serum (FCS) was then added, and cells were mechanically dissociated by 3 passages through a 10 ml pipette. Thereafter, cells were centrifuged at 180 x g, 10 min, 4°C on a layer of BSA (3.5%) in L15 medium. The cell pellet was re-suspended in culture medium containing Neurobasal medium with a 2% solution of B27 supplement, 2 mmol/liter of L-glutamine, 2% PS solution, 10 ng/ml of brain-derived neurotrophic factor (BDNF), 1 ng/ml of glial cell line-derived neurotrophic factor (GDNF), 4% heat-inactivated FCS, 1 g/L of glucose, 1 mM of sodium pyruvate, and 100 µM of non-essential amino acids. Cells were then seeded at a density of 80,000 cells/well in 96 well-plates pre-coated with poly-L-lysine, and maintained in a humidified incubator (37°C, 5% $CO_2$, 95% air). Half of the medium was replaced with fresh medium every 2 days.

On day 7 in culture, cells were pre-incubated for 1 hr with culture medium freshly spiked with known concentrations of DP, Pom or vehicle, and, thereafter, were exposed to an α-synuclein challenge (n = 6 per group). This challenge involved a 72 hr exposure to 250 nM α-synuclein (human recombinant α-synuclein 1–140 aa, rPetide, Watkinsville, GA, which a prior study reported contains >1.3 U endotoxin/mg of peptide, an endotoxin concentration incapable of producing any significant effects in terms of neurotoxicity or induction of ROS production [*Zhang et al., 2005*]). This

α-synuclein was previously prepared as a 4 µM solution and slowly shaken at 37°C for 72 hr in the dark to induce oligomerization. Thereafter, the cell culture supernatant was removed and frozen for later analysis and cells were washed in PBS.

Cellular assays: Cells were fixed in 4% paraformaldehyde in PBS (pH 7.3, 21°C, 20 min), washed twice in PBS, and then permeabilized. PBS containing 0.1% of saponin and 1% of FCS was added (21°C, 15 min) to eliminate non-specific binding. Thereafter, cells were then incubated with either (i) rabbit polyclonal anti tyrosine hydroxylase (TH) (diluted 1:2000 in PBS containing 1% FCS, 0.1% saponin, 21°C, 2 hr) to allow quantitation of dopaminergic neurons and neurites, or (ii) mouse monoclonal anti-OX-41 (diluted 1:500 in PBS containing 1% FCS and 0.1% of saponin, 21°C, 2 hr) to visualize microglia. After washing the sections, they were incubated with Alexa Fluor 488 goat anti-rabbit IgG (Jackson, 1:400) in PBS containing 1% FCS, 0.1% saponin (21°C) for 1 hr. Photomicrographs (20/ well x 10 magnification) were automatically acquired by ImageXpress (Molecular Devices, San Jose, CA) and were analyzed by Custom Module Editor (Molecular Devices).

## Statistical analysis
Data were evaluated across groups with one-way analysis of variance (ANOVA) followed by either a Tukey test or Dunnett test. The mean and standard error of the mean (S.E.M.) were calculated using Sigma Plot and Stat version 2.0 from Jandel Scientific, San Diego, CA. The Bonferroni correction was used for serial measurements. Bar graphs are presented as mean ± S.E.M. values. A value of $p<0.05$ or less was considered statistically significant.

## Acknowledgements
This research was supported in part by (i) the Intramural Research Program, National Institute on Aging, National Institutes of Health, USA, (ii) grants from (a) the Ministry of Science and Technology, Taiwan (MOST 104–2923-B-038–001-MY3 and MOST 108–2321-B-038–008), (b) DP2-107-21121-01-N-05, Taipei Medical University, Taipei, Taiwan, and (c) National Institutes of Health R56 AG057028, USA, and (iii) AevisBio Inc, in relation to co-author DSK.

## Additional information

### Competing interests
Weiming Luo, David Tweedie, Nigel H Greig: NHG, WL and DT are named inventors on patent 8,927,725 and have assigned all their rights to the NIH (US Government), and hence have no ownership of DP or other agents within US Patent 8,927,725. Dong Seok Kim: DSK is an employee of AevisBio, which is active in the area of neurological disorders, including brain injury. AevisBio does not stand to profit in any direct way from the publication of this article, but has an approved collaborative research and development agreement with NIA, NIH, and may or may not license technology from NIA, NIH, in the future. AevisBio had no input into the direction of the studies or interpretation of data in the manuscript. The other authors declare that no competing interests exist.

### Funding

| Funder | Grant reference number | Author |
|---|---|---|
| National Institutes of Health | AG000994 | Nigel H Greig |
| National Institutes of Health | AG057025 (R56) | Barry J Hoffer |
| Ministry of Science and Technology, Taiwan | MOST 104-2923-B-038-001-MY3 | Jia-Yi Wang |
| Ministry of Science and Technology, Taiwan | MOST 108-2321-B-038-008 | Jia-Yi Wang |
| Taipei Medical University | DP2-107-21121-01-N-05 | Jia-Yi Wang |
| AevisBio Inc | | Dong Seok Kim |

The funders had no role in study design, data collection and interpretation, or the decision to submit the work for publication.

### Author contributions
Chih-Tung Lin, Daniela Lecca, Ling-Yu Yang, Yoo-Jin Jung, Data curation, Formal analysis, Investigation; Weiming Luo, Resources, Synthetic chemistry of 3,6'-DP and Pom; Michael T Scerba, Resources, Writing - review and editing, Synthetic chemistry of 3,6-DT and Pom; David Tweedie, Data curation, Formal analysis, Investigation, Writing - review and editing; Pen-Sen Huang, Chih-Hao Yang, Data curation, Formal analysis; Dong Seok Kim, Formal analysis, Investigation, Writing - review and editing; Barry J Hoffer, Conceptualization, Supervision, Funding acquisition, Project administration, Writing - review and editing; Jia-Yi Wang, Conceptualization, Formal analysis, Supervision, Funding acquisition, Project administration, Writing - review and editing; Nigel H Greig, Conceptualization, Resources, Supervision, Funding acquisition, Writing - original draft, Project administration

### Author ORCIDs
David Tweedie (iD) http://orcid.org/0000-0002-8446-4544
Nigel H Greig (iD) https://orcid.org/0000-0002-3032-1468

### Ethics
Animal experimentation: This study was performed in strict accordance with the recommendations in the Guide for the Care and Use of Laboratory Animals of the National Institutes of Health. All animals were treated according to the International Guidelines for animal research, and the animal use protocols in this study were reviewed and approved by the Institutional Animal Care and Use Committee or Panel (IACUC/IACUP) of either the Intramural Research Program, National Institute on Aging, NIH (protocol No. 331-TGB-2021), or of Taipei Medical University [Protocol number: LAC-2015-0051]. All surgery was performed under appropriate anesthesia, and every effort was made to minimize suffering.

### Decision letter and Author response
Decision letter https://doi.org/10.7554/eLife.54726.sa1
Author response https://doi.org/10.7554/eLife.54726.sa2

## Additional files

### Supplementary files
• Transparent reporting form

### Data availability
All data generated or analysed during this study are included in the manuscript and supporting files. Source data files have been provided for all figures.

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
