## [Decision Letter]

**Acceptance summary:**

After careful examination by the reviewers of the revised version of the article, we feel that you article has addressed most of the concerns raised during the first round of review and provides interesting novel therapeutical approaches to TBI using an already approved FDA drug. The quality of the experiments and solidity of the data make this study an important contribution to the field.

**Decision letter after peer review:**

Thank you for submitting your article "3,6'-Dithiopomalidomide mitigates neuronal loss, inflammation and behavioral deficits in head injury and LPS challenge" for consideration by *eLife*. Your article has been reviewed by two peer reviewers, one of whom is a member of our Board of Reviewing Editors, and the evaluation has been overseen by Tadatsugu Taniguchi as the Senior Editor. The reviewers have opted to remain anonymous.

The reviewers have discussed the reviews with one another and the Reviewing Editor has drafted this decision to help you prepare a revised submission.

While both reviewers outlined the interest and the quality of your study, they raised some specific concerns, especially reviewer #2, that you should address in your revised version. It would in addition be important to frame your study in a broader context to explain the novelty of the work to non-specialist. We do hope that you will find the reviews constructive.

Reviewer #1:

In this manuscript by Lin et al., the authors explore the effect of 3,6'-Dithiopomalidomide (DP) on the deleterious effects of Traumatic Brain Injury (TBI). The laboratory had shown that pomalidomide (Pom), an immunomodulatory aminothalidomide analog that potently inhibits TNF-α production, had a positive effect on the outcome of cortical TBI in rats, and are hence now testing another analog that has similar effects on TNF-α but also lowers inflammation induced Cox2 and iNOS. Using a combination of well-designed and carefully quantified studies, the authors show that DP crosses the BBB and limits TNF-α production in vivo in response to LPS, DP iv 5 hours after the TBI alleviates behavioral symptoms, reduced contusion volume and improves functional outcome, with two doses and two stages tested. DP also reduces neuronal damage, caspase-3 induced apoptosis and microglial activation in the contusion region. Overall, in all tests, DP triggered more robust responses than Pom, especially at the dose of 0.5mg/kg. In addition, DP also selectively reduced the expression of autophagy markers and reduced astrogliosis, which was accompanied by a striking effect of DP pre-treatment in cell culture of nitrite, iNOS and COX2 levels. Finally, the authors show that Pom and DP treatments improves the survival and growth of cultured dopaminergic neurons exposed to α-synuclein, suggesting that it might be of interest in the treatment of Parkinson's disease.

Taken together, this is a well-designed and thorough study that systematically examines the effects of DP and compares them to Pom. Considering the previous work of the laboratory, it seems to more to constitute an interesting extension of the previous work than a fundamentally novel finding. The last experiments using α-synuclein are quite novel and appealing, as they raise the possibility that DP or Pom might be used in other diseases including PD. However, they lack in vivo support, which would require another major series of experiments. In this context, I cannot be fully supportive for publication in e*Life* and believe the study would be better suited for a more specialized journal.

Reviewer #2:

Manuscript entitled "3,6'-Dithiopomalidomide reduces neuronal loss, inflammation and behavioral deficits in brain injury and LPS challenge" by Lin.… Greig et al., is in the field of experimental traumatic brain injury where novel immunomodulatory agent DP was POM was investigated. DP and POM are both aminothalidomide analogues, and DP has not been studied before. POM has an extra amine group is the first bentzyl ring and DP has in addition two Sulphur at the second and 3rd ring. The advanced aspect of this study was that POM, was used as a control, to compare the finding to the previous study by the authors. Manuscript is well written, the aims are valid and the results support the conclusion. The field of the studies is very important since no drug treatment exists for TBI. The experiments are carefully conducted and the results are well and logically presented.

1) Number of animals for some of the behavioral experiments is rather low, e.g. n=5. Is there a reason for such a low n number for behavioral experiment where it is known that variation is rather high. Was power analysis conducted to show n=5 is enough?

2) Are the molecules directly modulating microglia polarization or is the effect on microglia derived from smaller lesions? One can study this with the later time point administration, where neuroprotection was not observed. It is known that e.g. after stroke microglia activation lasts for months. Therefore, the authors could study with the later time point administration whether molecules affect directly microglia polarization.

3) The effect on cytokine levels should be studies also at protein level (Figure 10). The qPCR experiments might not reflect protein levels. Also, it would be informative to include a wider analysis of different cytokines.

4) It remains unclear what is the direct target of the two compounds. Are they specific TNF-α synthesis inhibitors? Does POM and DP have same efficacy and affinity and what is their molecular target that they inhibit?

---

## [Author Response]

Reviewer #1:[…] Taken together, this is a well-designed and thorough study that systematically examines the effects of DP and compares them to Pom. Considering the previous work of the laboratory, it seems to more to constitute an interesting extension of the previous work than a fundamentally novel finding. The last experiments using α-synuclein are quite novel and appealing, as they raise the possibility that DP or Pom might be used in other diseases including PD. However, they lack in vivo support, which would require another major series of experiments. In this context, I cannot be fully supportive for publication in eLife and believe the study would be better suited for a more specialized journal.

We thank reviewer 1 for the very positive and constructive comments in relation to our study of the new experimental drug 3,6’-DithioPomalidomide (3,6-DP), and its comparison to the available drug Pomalidomide as a new treatment for TBI.

Our study represents the first detailed published study of 3,6-DP in animals, and compares it across cellular and animal studies to an equimolar dose of pomalidomide – an FDA approved drug for multiple myeloma – which has not been considered prior to us for its potential in neurodegenerative disorders. Our study hence provides two scientifically important advances: (i) it describes a new potential drug for TBI (a disorder for which there is no currently approved pharmacological treatment) and (ii) it compares it to a FDA approved drug with the potential for repositioning.

In relation to the reviewer’s final comment, the final study within the Results section of our manuscript relates to the actions of both drugs to mitigate α-synuclein-induced losses of dopaminergic neurons and their neurite extensions, and notably also demonstrates mitigation of microglial cell activation in primary cultures. We included this cellular α-synuclein study for multiple reasons. Most important, there is a broadly increasing agreement across epidemiological studies that TBI leads to an increased incidence of PD (see: Gardner et al., 2018; Crane et al., 2016 and multiple others). Notably, and as detailed by Acosta et al., 2015, α-synuclein provides an important pathological link between TBI and the later development of PD; this is reiterated in an extensive review by Delic et al., 2020, that has just appeared in the literature. Secondly, the application of α-synuclein to primary dopaminergic neuron and microglia cultures results in neuroinflammation, and provides a model that supplements the classical RAW 264.7 cells/LPS challenge model, and helps to transition it with the in vivo LPS and TBI studies in our manuscript.

The focus of our manuscript and studies is TBI, rather than the treatment of PD. We agree with the premise of reviewer 1 in that DP and Pom are interesting drugs that warrant evaluation in other neurological disorders that, like TBI, have an inflammatory component (such as PD). However, we view such an evaluation in an animal model of PD to be beyond the scope of this already comprehensive and detailed *eLife* submission (in this regard, one would need to select the appropriate PD model to evaluate, drug would be administered by a different route – as the i.v. route in our moderate TBI study would not be appropriate to PD treatment – and many other factors would require to be worked out – including drug synthesis in large amounts to support a chronic study (as DP is a new experimental drug – first synthesized by us – and not available anywhere else)). In the light of the reviewer’s comment – the initial lines of the Results section (subsection “DP and Pom mitigate α-synuclein-induced toxicity in primary dopaminergic cultures”) relating to our cellular dopaminergic neurons/microglia/α-synuclein study together with the Discussion section (eleventh paragraph) relating to α-synuclein have been substantially modified to stress the underpinning scientific rationale of our α-synuclein study and how it relates to the other studies within our manuscript.

Reviewer #2:[…] 1) Number of animals for some of the behavioral experiments is rather low, e.g. n=5. Is there a reason for such a low n number for behavioral experiment where it is known that variation is rather high. Was power analysis conducted to show n=5 is enough?

We evaluated 5 rats for each group, based on (i) the statistical significance achieved in our prior TBI studies (see, for example: Huang et al., 2018) and, (ii), a power analysis based on a study conducted by Charan and Kantharia, 2013, which suggests for any sample size, which keeps E (E = Total number of animals − Total number of groups) between 10 and 20 should be considered as adequate based on the ANOVA test. The E in our study is 24 (E=5*6-6) which is greater than 20, and means that adding further animals will not increase the chance of obtaining statistically significant results. Since our data showed clear statistical differences between Sham, TBI+Veh, Pom (0.5 and 0.1 mg/kg) and DP (0.1 and 0.5 mg/kg) groups, we did not add further animals to our behavioral experiments. As a general rule, we will add animals when there appears to be a clear trend that does not achieve statistical significance – or should an animal not be able to undertake a task (which would be noted in our manuscript, if occurring). In such a scenario we would include more animals according to 3R principle (reduction, replacement and refinement) for ethical use of animals.

In the light of the reviewer’s comment, we have modified the appropriate section within the Materials and methods section to indicate “……(n=5 per group, based on both prior studies (Huang et al., 2018; Yang et al., 2016, 2015) and a power analysis (Charan and Kantharia, 2013).”

2) Are the molecules directly modulating microglia polarization or is the effect on microglia derived from smaller lesions? One can study this with the later time point administration, where neuroprotection was not observed. It is known that e.g. after stroke microglia activation lasts for months. Therefore, the authors could study with the later time point administration whether molecules affect directly microglia polarization.

In revised Figure 10, we now provide data where we examined the expression of key genes associated with M1 (TNF-α, IL-1β, IL-6) or M2 microglia (arginase-1, IL-4 and IL-10 (new panels E, F and G) phenotypes (subsection “Post-injury administration of DP or Pom significantly attenuated TBI-induced elevations in mRNA expression levels of pro-inflammatory cytokines”). In addition, iNOS data is available in Figure 11. Our data indicates that TBI upregulates genes associated with M1 microglia and down-regulates genes associated with M2 microglia. Intravenous administration of Pom (0.5 mg/kg) and, in particular, DP (0.5 mg/kg) reversed microglia polarization caused by TBI – by down-regulating M1 and upregulating M2 genes. Notably, protein levels of TNF-α, IL-1β and IL-6 have now also been measured, and mirror changes in mRNA expression (new panels H, I and J in Figure 10).

To further answer the reviewer’s question on microglia activation, we also revised Figure 9 by adding new panels D and E- in which we analyzed the proportion of microglia with an activated morphology (revised Figure 9).

3) The effect on cytokine levels should be studies also at protein level (Figure 10). The qPCR experiments might not reflect protein levels. Also, it would be informative to include a wider analysis of different cytokines.

We agree, and performed additional experiments to measure cytokine protein levels (new panels H, I and J in revised Figure 10). Protein changes mirror those of mRNA. As noted above, in our response to comment 2, we have measured mRNA levels of three markers of anti-inflammatory M2-like microglia (arginase-1, IL4 and IL-10 – new panels D, E and F within Figure 10). Furthermore, we have now evaluated resting vs. activated microglia in Figure 9 (new panels D and E)

4) It remains unclear what is the direct target of the two compounds. Are they specific TNF-α synthesis inhibitors? Does POM and DP have same efficacy and affinity and what is their molecular target that they inhibit?

The reviewer poses an interesting question, as to the specific target of DP and POM. In large part and as a background, the ‘drug development’ process can be split into ‘target-based’ vs. ‘phenotypic’ screening and subsequent development. The emerging consensus is that multiple TBI-associated cascades are triggered in parallel and, if true, the multifactorial nature of TBI would make the discovery of a single effective ‘mechanism-targeted’ drug improbable. To counter this, phenotypic screens can be used in drug development to identify compounds to mitigate a critical step(s) associated with TBI, such as neuroinflammation, and such a phenotypic screen is largely agnostic to any specific mechanism(s) underpinning this event. We selected DP from a broad number of novel Pom analogues that we synthesized and screened (Beedie et al., 2016; Luo et al., 2018) via phenotypic-screening to identify agents that mitigated inflammation on the Pom backbone. Key amongst these screens is the RAW 264.7 cell/LPS screen, which is known to induce the cells to generate and release TNF-α and reactive nitrogen species. TNF-α is hence considered a primary target of Pom and related drugs, as noted in the Discussion; such agents enhance mRNA degradation of TNF-α post-transcriptionally and, thereby, lower TNF-α generation (Discussion, fourth paragraph). As also noted in the Discussion, either through lowered TNF-α or via direct actions on other cytokines, DP appears to have a range of potentially useful actions (that seems to be wider than that of Pom). This advantage of phenotypic drug screening may account for it being associated with the highest number of first-in-class drug approvals by the FDA between 1999 and 2008 (Swinney and Anthony, 2011).